# Convergence Rate Analysis of the AdamW-Style Shampoo: Unifying One-Sided and Two-Sided Preconditioning

Huan Li [1]  Yiming Dong [2]  Zhouchen Lin [2]

## Abstract

This paper studies the AdamW-style Shampoo optimizer, an effective implementation of the classical Shampoo that notably won the external tuning track of the AlgoPerf neural network training algorithm competition (Kasimbeg et al., 2025). Our analysis unifies one-sided and two-sided preconditioning and establishes the convergence rate $\frac{1}{K}\sum_{k=1}^{K}\mathbb{E}\left[\|\nabla f(\mathbf{X}_k)\|_*\right] \leq \mathcal{O}(\frac{\sqrt{m+n}C}{K^{1/4}})$ measured by nuclear norm, where $K$ represents the iteration number, $(m, n)$ denotes the size of matrix parameters, and $C$ matches the constant in the optimal convergence rate of SGD. Theoretically, we have $\|\nabla f(\mathbf{X})\|_F \leq \|\nabla f(\mathbf{X})\|_* \leq \sqrt{\min\{m,n\}}\|\nabla f(\mathbf{X})\|_F$, supporting that our convergence rate can be considered to be analogous to the optimal $\frac{1}{K}\sum_{k=1}^{K}\mathbb{E}\left[\|\nabla f(\mathbf{X}_k)\|_F\right] \leq \mathcal{O}(\frac{C}{K^{1/4}})$ convergence rate of SGD in the ideal case of $\|\nabla f(\mathbf{X})\|_* = \Theta(\sqrt{\min\{m,n\}})\|\nabla f(\mathbf{X})\|_F$ and balanced $m$ and $n$.

## 1. Introduction

Adaptive gradient methods have become the predominant optimizers for training deep neural networks, especially in large language models. The development of adaptive gradient algorithms has followed two distinct lineages: diagonal preconditioning and non-diagonal preconditioning. The former has developed through the progression of AdaGrad (Duchi et al., 2011; McMahan & Streeter, 2010), RMSProp (Tieleman & Hinton, 2012), Adam (Kingma & Ba, 2015), and finally AdamW (Loshchilov & Hutter, 2019), which have served as the de facto optimizer for training deep networks over the past decade. The latter lineage, originating from full-matrix AdaGrad (Duchi et al., 2011) and advancing to methods such as K-FAC (Martens & Grosse, 2015), Shampoo (Gupta et al., 2018), SOAP (Vyas et al., 2025), and Muon (Jordan et al., 2024), now demonstrates the potential to outperform its diagonal counterparts.

Diagonally preconditioned methods typically employ coordinate-wise scaling to the gradient. For instance, Ada-Grad treats the network's parameters as a single high-dimensional vector and updates them according to the following procedure

$$\mathbf{x}_{k+1} = \mathbf{x}_k - \eta\Lambda_k^{-1/2}\mathbf{g}_k, \quad \Lambda_k = \text{diag}\left(\sum_{t=1}^{k}\text{diag}(\mathbf{g}_t\mathbf{g}_t^T)\right),$$

where $\Lambda_k$ is a diagonal preconditioning matrix whose entries are the element-wise sum of squared historical gradients.

In contrast, non-diagonally preconditioned methods exploit the inherent matrix structure of parameters in deep networks. For example, the Shampoo optimizer operates according to the following formulation with two-sided preconditioning

$$\mathbf{X}_{k+1} = \mathbf{X}_k - \eta\mathbf{L}_k^{-1/4}\mathbf{G}_k\mathbf{R}_k^{-1/4},$$
$$\mathbf{L}_k = \sum_{t=1}^{k}\mathbf{G}_t\mathbf{G}_t^T, \quad \mathbf{R}_k = \sum_{t=1}^{k}\mathbf{G}_t^T\mathbf{G}_t,$$

where the gradient $\mathbf{G}_k \in \mathbb{R}^{m\times n}$ is a matrix and $\mathbf{L}_k \in \mathbb{R}^{m\times m}$ and $\mathbf{R}_k \in \mathbb{R}^{n\times n}$ are non-diagonal preconditioners consisting of sum of historical gradient outer products. It can be regarded as using the Kronecker product $\mathbf{R}_k^{1/2} \otimes \mathbf{L}_k^{1/2}$ to approximate the full-matrix AdaGrad preconditioner $\sum_{t=1}^{k}\mathbf{g}_t\mathbf{g}_t^T$, where $\mathbf{g}_t = \text{vec}(\mathbf{G}_t)$.

A key advantage of non-diagonally preconditioned methods is their ability to capture the cross-parameter correlations within the gradient, thereby yielding a more informed search direction and potentially superior convergence compared to diagonal approaches. Recently, an implementation based on the distributed Shampoo (Anil et al., 2020; Shi et al., 2023) won the external tuning track of the AlgoPerf neural network training algorithm competition (Kasimbeg et al., 2025), demonstrating that non-diagonally preconditioned

[1]College of Artificial Intelligence, Nankai University, Tianjin, China. [2]State Key Lab of General AI, School of Intelligence Science and Technology, Peking University, Beijing, China. Correspondence to: Huan Li <lihuanss@nankai.edu.cn>, Zhouchen Lin <zlin@pku.edu.cn>.

*Proceedings of the 43rd International Conference on Machine Learning*, Seoul, South Korea. PMLR 306, 2026. Copyright 2026 by the author(s).

---

**Algorithm 1** AdamW-style Shampoo

Hyper parameters: $\eta, \theta, \beta, \lambda, \varepsilon$, positive $p, q$ with $\frac{1}{p}+\frac{1}{q}=1$.

Denote $\mathbf{L}_{k,\varepsilon}^{\pm\frac{1}{\infty}} = \mathbf{I}_m$ and $\mathbf{R}_{k,\varepsilon}^{\pm\frac{1}{\infty}} = \mathbf{I}_n$.

Initialize $\mathbf{X}_1, \mathbf{M}_0 = \mathbf{0}, \mathbf{L}_0 = \mathbf{0}, \mathbf{R}_0 = \mathbf{0}$.

**for** $k = 1, 2, \cdots, K$ **do**

$\quad\mathbf{G}_k = \text{GradOracle}(\mathbf{X}_k)$

$\quad\mathbf{M}_k = \theta\mathbf{M}_{k-1} + (1 - \theta)\mathbf{G}_k$

$\quad\mathbf{L}_k = \beta\mathbf{L}_{k-1} + (1 - \beta)\mathbf{G}_k\mathbf{G}_k^T$

$\quad\mathbf{R}_k = \beta\mathbf{R}_{k-1} + (1 - \beta)\mathbf{G}_k^T\mathbf{G}_k$

$\quad\mathbf{L}_{k,\varepsilon} = \mathbf{L}_k + \varepsilon\mathbf{I}_m, \quad \mathbf{R}_{k,\varepsilon} = \mathbf{R}_k + \varepsilon\mathbf{I}_n$

$\quad\mathbf{X}_{k+1} = (1 - \lambda\eta)\mathbf{X}_k - \eta\mathbf{L}_{k,\varepsilon}^{-\frac{1}{2p}}\mathbf{M}_k\mathbf{R}_{k,\varepsilon}^{-\frac{1}{2q}}$

**end for**

---

training algorithms can outperform currently popular diagonal preconditioning methods, such as Adam. The winner implementation achieved significantly accelerated training, with an average speedup of $28\%$ over the NAdamW (Dozat, 2016) baseline across eight deep learning workloads. Algorithm 1 presents the core characteristics of the Shampoo implemented in (Anil et al., 2020; Shi et al., 2023) in a non-distributed manner, including the exponential moving average of the first and second moment matrices, two-sided preconditioning with a tunable exponent, and decoupled weight decay.

Theoretically, convergence of diagonally preconditioned methods has been extensively studied (Défossez et al., 2022; Shi et al., 2020; Li et al., 2025b; Zhang et al., 2022; Hong & Lin, 2024; Li et al., 2023; 2025a). For non-diagonal methods, Muon represents the first method to receive a rigorous convergence analysis for nonconvex optimization (Li & Hong, 2025; Kim & hwan Oh, 2026; Shen et al., 2025; Chen et al., 2026; Sato et al., 2025). Analyses of other optimizers in this class, such as Shampoo, have largely been confined to convex settings. For example, Gupta et al. (2018) established the regret bound of Shampoo within the online convex optimization framework, Xie et al. (2025) provided a unified convergence analysis including full-matrix AdaGrad and *one-sided* variant of Shampoo for convex problems, where the update $\mathbf{L}_k^{-1/4}\mathbf{G}_k\mathbf{R}_k^{-1/4}$ in the original Shampoo is replaced by $\mathbf{L}_k^{-1/2}\mathbf{G}_k$, An et al. (2025) proposed ASGO, effectively equivalent to *one-sided* Shampoo, and studied its convergence for convex programming.

To the best of our knowledge, (Xie et al., 2026) appears to be the only work prior to ours that establishes the convergence of Shampoo in the nonconvex setting. However, their study is limited to the *one-sided* variant of Shampoo in the *AdaGrad-style* and *RMSProp-style*, and does not address the more complex, yet more commonly used, two-sided preconditioning. Furthermore, their analysis does not incorporate momentum or decoupled weight decay. While

other works (Feinberg et al., 2023; Morwani et al., 2025; Eschenhagen et al., 2025; Lin et al., 2026) have explored Shampoo from different perspectives, none have provided a convergence guarantee for the nonconvex case.

In this paper, we study the AdamW-style Shampoo presented in Algorithm 1, which provides a unified treatment of two-sided ($p, q < +\infty$) and one-sided ($p = 1, q = +\infty$ or $p = +\infty, q = 1$) preconditioning. This formulation represents the most effective practical implementation of Shampoo (Anil et al., 2020; Shi et al., 2023), as demonstrated in (Kasimbeg et al., 2025).

### 1.1. Contributions

This paper establishes the following convergence rate of Algorithm 1 for nonconvex programming

$$\frac{1}{K}\sum_{k=1}^{K}\mathbb{E}\left[\|\nabla f(\mathbf{X}_k)\|_*\right] \leq \mathcal{O}\left(\sqrt{m+n}\right) \times$$

$$\max\left\{\sqrt[4]{\frac{\sigma^2 L\left(f(\mathbf{X}_1) - f^*\right)}{K}}, \sqrt{\frac{L\left(f(\mathbf{X}_1) - f^*\right)}{K}}\right\}$$

while ensuring $\|\mathbf{X}_k\|_{op} < \frac{1}{\lambda}$ for all $k = 1, 2, \cdots, K$, where the notations can be found in Section 1.2. As a comparison, the classical convergence rate of SGD is (Bottou et al., 2018)

$$\frac{1}{K}\sum_{k=1}^{K}\mathbb{E}\left[\|\nabla f(\mathbf{X}_k)\|_F\right] \leq \mathcal{O}\left(\sqrt[4]{\frac{\sigma^2 L\left(f(\mathbf{X}_1) - f^*\right)}{K}}\right), \quad (1)$$

which matches the lower bound of nonconvex stochastic optimization (Arjevani et al., 2023). Since Frobenius norm and nuclear norm satisfy

$$\|\nabla f(\mathbf{X})\|_F \leq \|\nabla f(\mathbf{X})\|_* \leq \sqrt{\min\{m, n\}}\|\nabla f(\mathbf{X})\|_F,$$

our convergence rate also aligns with the same lower bound with respect to all the coefficients in the ideal case of $\|\nabla f(\mathbf{X})\|_* = \Theta(\sqrt{\min\{m, n\}})\|\nabla f(\mathbf{X})\|_F$ and balanced $m$ and $n$, which is verified empirically on real training of GPT-2 in our experiment.

### 1.2. Problem Settings, Notations, and Assumptions

We study the following nonconvex problem with matrix parameters in this paper

$$\min_{\mathbf{X}\in\mathbb{R}^{m\times n}} f(\mathbf{X}), \quad (2)$$

where $f(\mathbf{X}) = \mathbb{E}_{\zeta\in\mathcal{P}}[f(\mathbf{X};\zeta)]$ and $\zeta$ is the sample drawn from the data distribution $\mathcal{P}$.

We denote vectors by lowercase bold letters and matrices by uppercase bold letters. We use $\mathbf{I}_m$ for the identity matrix in $\mathbb{R}^{m\times m}$. For vectors, denote $\|\cdot\|$ as the $\ell_2$ Euclidean

norm. For matrices, denote $\|\cdot\|_F$, $\|\cdot\|_{op}$, and $\|\cdot\|_*$ as the Frobenius norm, spectral norm (largest singular value), and nuclear norm (sum of singular values), respectively. The trace of a square matrix is written as $\mathrm{tr}(\cdot)$. Denote $\mathcal{F}_k = \sigma(\mathbf{G}_1, \mathbf{G}_2, \cdots, \mathbf{G}_k)$ to be the sigma field of the stochastic gradients up to $k$, denote $\mathbb{E}_{\mathcal{F}_k}[\cdot]$ as the expectation with respect to $\mathcal{F}_k$ and $\mathbb{E}_k[\cdot|\mathcal{F}_{k-1}]$ the conditional expectation with respect to $\mathbf{G}_k$ given $\mathcal{F}_{k-1}$. For brevity, $\mathbb{E}_{\mathcal{F}_K}[\cdot]$ will be denoted as $\mathbb{E}[\cdot]$. Let $f^*$ denote the lower bound of $f(\mathbf{X})$. Finally, denote the singular values of $\mathbf{A} \in \mathbb{R}^{m \times n}$ by $\sigma_1(\mathbf{A}), \cdots, \sigma_r(\mathbf{A})$ in a nonincreasing order with $r = \min\{m, n\}$.

We make the following assumptions throughout this paper:

1. Smoothness:
   $\|\nabla f(\mathbf{Y}) - \nabla f(\mathbf{X})\|_F \leq L\|\mathbf{Y} - \mathbf{X}\|_F, \forall \mathbf{X}, \mathbf{Y},$

2. Unbiased estimator: $\mathbb{E}_k\left[\mathbf{G}_k\big|\mathcal{F}_{k-1}\right] = \nabla f(\mathbf{X}_k),$

3. Bounded row-wise and column-wise second central moment matrices:
   $\mathbb{E}_k\left[\left(\mathbf{G}_k - \nabla f(\mathbf{X}_k)\right)\left(\mathbf{G}_k - \nabla f(\mathbf{X}_k)\right)^T\big|\mathcal{F}_{k-1}\right] \preceq \Sigma_L,$
   $\mathbb{E}_k\left[\left(\mathbf{G}_k - \nabla f(\mathbf{X}_k)\right)^T\left(\mathbf{G}_k - \nabla f(\mathbf{X}_k)\right)\big|\mathcal{F}_{k-1}\right] \preceq \Sigma_R$
   with symmetric positive semidefinite matrices $\Sigma_L$ and $\Sigma_R$.

The first two assumptions are identical to the standard assumptions used in the analysis of SGD, while the third assumption is more restrictive than that in SGD analysis. In fact, from the third assumption, it readily follows that

$$\mathbb{E}_k\left[\|\mathbf{G}_k - \nabla f(\mathbf{X}_k)\|_F^2\big|\mathcal{F}_{k-1}\right] \leq \frac{\mathrm{tr}(\Sigma_L) + \mathrm{tr}(\Sigma_R)}{2} \equiv \sigma^2, \tag{3}$$

which is the standard bounded variance assumption in SGD analysis.

## 2. Convergence Rate of the AdamW-Style Shampoo

Based on Assumptions 1-3, we establish the convergence rate of Algorithm 1 in the following theorem. Due to the definitions of $\mathbf{L}_{k,\varepsilon}$ and $\mathbf{R}_{k,\varepsilon}$, condition (4) always holds for $\hat{\varepsilon} = \varepsilon$.

**Theorem 2.1.** *Suppose that Assumptions 1-3 and condition*

$$\mathbf{L}_{k,\varepsilon} \succeq \hat{\varepsilon}\mathbf{I}_m, \quad \mathbf{R}_{k,\varepsilon} \succeq \hat{\varepsilon}\mathbf{I}_n \tag{4}$$

*hold for some $\hat{\varepsilon} \geq \varepsilon$. Let $\hat{\sigma}^2 = \max\left\{\sigma^2, \frac{L(f(\mathbf{X}_1)-f^*)}{K\gamma^2}\right\}$ with any $\gamma \in (0,1]$, $\frac{1}{p} + \frac{1}{q} = 1$, $1 - \theta = \sqrt{\frac{L(f(\mathbf{X}_1)-f^*)}{K\hat{\sigma}^2}}$, $\theta \leq \beta \leq \sqrt{\theta}$, $\varepsilon = \frac{\tau\hat{\sigma}^2}{m+n}$ with any $\tau \leq 1$ being the hyperparameter to make $\varepsilon$ small in practice, $\eta = \sqrt{\frac{\hat{\varepsilon}(f(\mathbf{X}_1)-f^*)}{4LK\hat{\sigma}^2}}$, $\lambda \leq \frac{1}{\sqrt{1152\hat{\varepsilon}}K^{3/4}}\sqrt[4]{\frac{L^3\hat{\sigma}^2}{f(\mathbf{X}_1)-f^*}}$, and $\|\mathbf{X}_1\|_{op} \leq \sqrt{\frac{\hat{\varepsilon}K(f(\mathbf{X}_1)-f^*)}{L\hat{\sigma}^2}}$.*

*Then for Algorithm 1, we have $\|\mathbf{X}_k\|_{op} < \frac{1}{\lambda}$ for all $k = 1, 2, \cdots, K$ and*

$$\frac{1}{K}\sum_{k=1}^{K}\mathbb{E}\left[\|\nabla f(\mathbf{X}_k)\|_*\right] \leq \left(8\sqrt{m+n} + \frac{119\hat{\sigma}}{\sqrt{\hat{\varepsilon}}}\right) \times$$
$$\max\left\{\sqrt[4]{\frac{\sigma^2 L(f(\mathbf{X}_1)-f^*)}{K}}, \sqrt{\frac{L(f(\mathbf{X}_1)-f^*)}{K\gamma}}\right\}. \tag{5}$$

*In the worst case, when $\hat{\varepsilon} = \varepsilon$, we have*

$$\frac{1}{K}\sum_{k=1}^{K}\mathbb{E}\left[\|\nabla f(\mathbf{X}_k)\|_*\right] \leq 127\sqrt{\frac{m+n}{\tau}} \times$$
$$\max\left\{\sqrt[4]{\frac{\sigma^2 L(f(\mathbf{X}_1)-f^*)}{K}}, \sqrt{\frac{L(f(\mathbf{X}_1)-f^*)}{K\gamma}}\right\}. \tag{6}$$

*Furthermore, when $\tau = 1$, we achieve the best theoretical convergence rate*

$$\frac{1}{K}\sum_{k=1}^{K}\mathbb{E}\left[\|\nabla f(\mathbf{X}_k)\|_*\right] \leq 127\sqrt{m+n} \times$$
$$\max\left\{\sqrt[4]{\frac{\sigma^2 L(f(\mathbf{X}_1)-f^*)}{K}}, \sqrt{\frac{L(f(\mathbf{X}_1)-f^*)}{K\gamma}}\right\}. \tag{7}$$

**Discussions on $\varepsilon$ and $\hat{\varepsilon}$.** In practice, the parameter $\varepsilon$ is typically set to a very small value, for example, $10^{-12}$ as used in (Shi et al., 2023). On the other hand, in modern large language models, the dimensions of weight matrices optimized by non-diagonally preconditioned methods such as Shampoo/SOAP/Muon are not exceptionally large. For instance, in GPT-3 with 175 bililion parameters, the QKV projection matrices have dimensions $m = n = 12288$, while the weight matrices in the feed-forward network layer have dimensions $(m, n) = (12288, 49152)$ or $(49152, 12288)$. Consequently, the quantity $\frac{\hat{\sigma}^2}{m+n}$ is several orders of magnitude larger than the $\varepsilon$ used in practice. To reconcile this discrepancy, we introduce the scaling factor $\tau$ into the setting of $\varepsilon$, aligning the analysis with practical configurations. This adjustment, however, leads to a slower convergence rate, as shown in (6), which depends explicitly on $\tau$. The impractical setting $\tau = 1$ yields the best convergence rate given in (7).

To bridge this gap between theory and practice, we further introduce condition (4). Informally, the preconditioners $\mathbf{L}_k$ and $\mathbf{R}_k$ can be regarded as approximating $\mathbb{E}\left[\mathbf{G}\mathbf{G}^T\right]$ and $\mathbb{E}\left[\mathbf{G}^T\mathbf{G}\right]$, respectively. In the training of modern large language models, empirical evidence indicates that the gradient norm remains of $\mathcal{O}(1)$ (Wen et al., 2026, Figure 7). Consequently, Lemma 2.2 suggests that condition (4) is reasonable with $\hat{\varepsilon} = \mathcal{O}(\frac{1}{m+n})$. As illustrated in Figure 3 in Appendix E, this condition is empirically satisfied during

GPT-2 training for a moderate value of $\hat\varepsilon$, which remains orders of magnitude larger than $\varepsilon$. With condition (4), our derived convergence rate (5) depends only on $\hat\varepsilon$, rather than $\varepsilon$ or $\tau$. When $\hat\varepsilon \geq \frac{\hat\sigma^2}{m+n}$, convergence rate (5) matches (7), even for arbitrarily small $\tau$. This represents a trade-off between theoretical guarantees and practical behavior. From the perspective of worst-case theoretical bounds, (7) yields the optimal convergence rate, and condition (4) can be removed since it is always satisfied with $\hat\varepsilon = \varepsilon$. From a practical standpoint, (5) better explains why setting $\varepsilon$ to an extremely small value does not hinder fast convergence in real-world scenarios.

**Lemma 2.2.** *When each element of $\mathbf{G} \in \mathbb{R}^{m \times n}$ is generated from Gaussian distribution with mean $\mu$ and variance $\xi^2$ independently, we have*

$$\mathbb{E}\left[\mathbf{G}\mathbf{G}^T\right] \succeq n\xi^2 \mathbf{I}_m = \frac{\xi^2}{m(\xi^2+\mu^2)}\mathbb{E}\left[\|\mathbf{G}\|_F^2\right]\mathbf{I}_m,$$

$$\mathbb{E}\left[\mathbf{G}^T\mathbf{G}\right] \succeq m\xi^2 \mathbf{I}_n = \frac{\xi^2}{n(\xi^2+\mu^2)}\mathbb{E}\left[\|\mathbf{G}\|_F^2\right]\mathbf{I}_n.$$

**Optimality of our convergence rate**. Comparing the optimal convergence rate (1) of SGD with our best theoretical convergence rate (7), we observe that our result employs the nuclear norm and introduces an additional factor of $\sqrt{m+n}$. Denoting $\sigma_1, \sigma_2, \cdots, \sigma_r$ to be the singular values of $\nabla f(\mathbf{X})$ with $r = \min\{m,n\}$, the Frobenius norm and nuclear norm satisfy

$$\|\nabla f(\mathbf{X})\|_F = \sqrt{\sum_{i=1}^r \sigma_i^2} \leq \sum_{i=1}^r \sigma_i = \|\nabla f(\mathbf{X})\|_*,$$

$$\|\nabla f(\mathbf{X})\|_* = \sum_{i=1}^r \sigma_i \leq \sqrt{r\sum_{i=1}^r \sigma_i^2} = \sqrt{r}\|\nabla f(\mathbf{X})\|_F,$$

which means that our convergence rate also aligns with the lower bound in nonconvex stochastic optimization (Arjevani et al., 2023) in the ideal case of $\|\nabla f(\mathbf{X})\|_* = \Theta(\sqrt{\min\{m,n\}})\|\nabla f(\mathbf{X})\|_F$ and balanced $m$ and $n$, which is verified empirically on real training of GPT-2, as demonstrated in Figure 2 in Appendix E.

According to the recent work (Jiang et al., 2025), the lower bound for AdaGrad is $\mathbb{E}\left[\min_{1 \leq k \leq K} \|\nabla f(\mathbf{x}_k)\|_1\right] = \Theta\left(\frac{\sqrt{d}}{K^{1/4}}\sqrt[4]{\sigma^2 L(f(\mathbf{x}_1)-f^*)}\right)$ for a constructed function $f(\mathbf{x})$ under standard assumptions of $L$-smoothness and $\sigma^2$-bounded variance (3), where $d$ denotes the dimension of the variable. Consider the following simplified scenario: during the iteration process, the left and right singular matrices of $\mathbf{G}_k$ remain unchanged with $\mathbf{U}\Sigma_k\mathbf{V}^T$ being its compact singular value decomposition (SVD), then Shampoo is equivalent to running AdaGrad on the singular values in the sense of $\mathbf{L}_k^{-1/4}\mathbf{G}_k\mathbf{R}_k^{-1/4} =$

$\mathbf{U}\left(\sum_{t=1}^k \Sigma_t^2\right)^{-1/4}\Sigma_k\left(\sum_{t=1}^k \Sigma_t^2\right)^{-1/4}\mathbf{V}^T$. Since the AdamW-style Shampoo further extends Shampoo, and given the unavoidable $\sqrt{d}$ dependence in AdaGrad's lower bound, we conjecture that the convergence rate derived in our paper is sharp and that the factor $\sqrt{m+n}$ cannot be eliminated.

**Unifying two-sided and one-sided preconditioning**. When $p, q < +\infty$, Algorithm 1 employs two-sided preconditioning; for instance, setting $p = q = 2$ recovers the original Shampoo update proposed in (Gupta et al., 2018). If either $p$ or $q$ is infinite, the algorithm reduces to the one-sided preconditioning analyzed in (Xie et al., 2025; An et al., 2025; Xie et al., 2026). Our analysis framework permits any positive values $p$ and $q$ (including infinite ones) satisfying $\frac{1}{p} + \frac{1}{q} = 1$. Empirically, Anil et al. (2020) and Shi et al. (2023) observed that treating the exponents $p$ and $q$ as tunable hyperparameters can lead to improved performance. Intuitively, the left preconditioning $\mathbf{L}^{-\frac{1}{2}}\mathbf{M}$ captures correlations within each column of $\mathbf{M}$, while the right preconditioning $\mathbf{M}\mathbf{R}^{-\frac{1}{2}}$ captures correlations within each row of $\mathbf{M}$. Two-sided preconditioning $\mathbf{L}^{-\frac{1}{2p}}\mathbf{M}\mathbf{R}^{-\frac{1}{2q}}$ combines these advantages and captures correlations within both rows and columns of $\mathbf{M}$.

**AdamW-style Shampoo v.s. AdamW itself**. Li et al. (2025a) established the convergence rate

$$\frac{1}{K}\sum_{k=1}^K \mathbb{E}\left[\|\nabla f(\mathbf{x}_k)\|_1\right]$$
$$\leq \mathcal{O}\left(\frac{\sqrt{d}}{K^{1/4}}\sqrt[4]{\sigma^2 L(f(\mathbf{x}_1)-f^*)} + \sqrt{\frac{dL(f(\mathbf{x}_1)-f^*)}{K}}\right)$$

for AdamW while ensuring $\|\mathbf{x}_k\|_\infty < \frac{1}{\lambda}$ for all $k = 1, 2, \cdots, K$, where $d$ is the dimension. Comparing with (7), we observe that AdamW-style Shampoo employs the nuclear norm in its convergence rate and spectral norm in its implicit bias, which correspond to the $\ell_1$ and $\ell_\infty$ norms applied to the singular values, respectively. Consequently, AdamW-style Shampoo can be interpreted as achieving theoretical behavior analogous to AdamW, but in the space of singular values.

**AdamW-style Shampoo v.s. Muon**. Let $\mathbf{U}_k\Sigma_k\mathbf{V}_k^T$ be the compact singular value decomposition (SVD) of $\mathbf{M}_k$. From the identities $\left(\mathbf{M}_k\mathbf{M}_k^T\right)^{\frac{1}{2p}} = \mathbf{U}_k\Sigma_k^{\frac{1}{p}}\mathbf{U}_k^T$ and $\left(\mathbf{M}_k^T\mathbf{M}_k\right)^{\frac{1}{2q}} = \mathbf{V}_k\Sigma_k^{\frac{1}{q}}\mathbf{V}_k^T$, the update $\mathbf{L}_{k,\varepsilon}^{-\frac{1}{2p}}\mathbf{M}_k\mathbf{R}_{k,\varepsilon}^{-\frac{1}{2q}}$ in AdamW-style Shampoo can be written equivalently as[1]

$$\mathbf{L}_{k,\varepsilon}^{-\frac{1}{2p}}\left(\mathbf{M}_k\mathbf{M}_k^T\right)^{\frac{1}{2p}}\mathbf{U}_k\mathbf{V}_k^T\left(\mathbf{M}_k^T\mathbf{M}_k\right)^{\frac{1}{2q}}\mathbf{R}_{k,\varepsilon}^{-\frac{1}{2q}}.$$

In an informal sense, $\mathbf{M}_k$, $\mathbf{L}_k$, and $\mathbf{R}_k$ can be interpreted as approximations to the first moment matrix $\mathbb{E}\left[\mathbf{G}\right]$,

---

[1]Concurrent with this work, we noticed that Eschenhagen et al. (2026) discovered a similar formula.

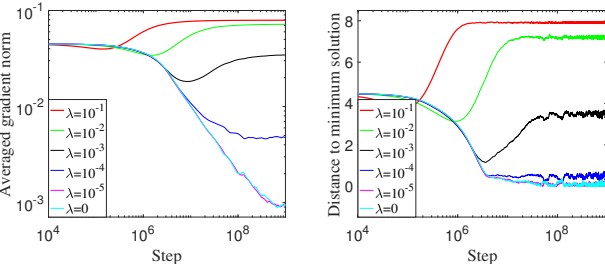

*Figure 1.* Illustrations of $\frac{1}{k}\sum_{t=1}^{k}\|\nabla f(\mathbf{X}_t)\|_F$ (left) and $\|\mathbf{X}_k - \mathbf{X}^*\|_F$ (right) over steps on the toy example (8).

the row-wise second raw moment matrix $\mathbb{E}\left[\mathbf{G}\mathbf{G}^T\right]$, and the column-wise second raw moment matrix $\mathbb{E}\left[\mathbf{G}^T\mathbf{G}\right]$, respectively. Noting the decomposition $\mathbb{E}\left[\mathbf{G}\mathbf{G}^T\right] = (\mathbb{E}[\mathbf{G}])(\mathbb{E}[\mathbf{G}])^T + \mathbb{E}\left[(\mathbf{G}-\mathbb{E}[\mathbf{G}])(\mathbf{G}-\mathbb{E}[\mathbf{G}])^T\right]$, the quantities $\mathbf{L}_{k,\varepsilon}^{-\frac{1}{2p}}\left(\mathbf{M}_k\mathbf{M}_k^T\right)^{\frac{1}{2p}}$ and $\left(\mathbf{M}_k^T\mathbf{M}_k\right)^{\frac{1}{2q}}\mathbf{R}_{k,\varepsilon}^{-\frac{1}{2q}}$ can therefore be regarded as the row-wise and column-wise signal-to-total-energy ratio (STR) matrices, respectively. This is a conceptual extension from scalar notations in the original Adam to matrices (Kingma & Ba, 2015). Consequently, AdamW-style Shampoo can be viewed as an STR preconditioned variant of Muon. This relationship is analogous to that between AdamW and SignSGD (Orvieto & Gower, 2025).

**Restricted weight decay parameter**. Our theory requires the weight decay parameter $\lambda$ to be sufficiently small. To demonstrate that such an upper bound is necessary for convergence, we follow (Li et al., 2025a) to consider a simple stochastic convex problem

$$f(\mathbf{X}) = \frac{\|\mathbf{X}-\mathbf{X}^*\|_F^2}{200} \quad \text{with} \quad \mathbf{X}^* = \begin{bmatrix} 4, & 4 \\ 4, & 4 \end{bmatrix}. \quad (8)$$

The stochastic gradient is given by

$$\mathbf{G}(\mathbf{X}) = \begin{cases} \mathbf{X}-\mathbf{X}^*-\mathbf{A}, & \text{with probability } 0.1, \\ -\frac{1}{10}(\mathbf{X}-\mathbf{X}^*-\frac{10}{9}\mathbf{A}), & \text{with probability } 0.9. \end{cases}$$

We initialize $\mathbf{X}_1 = \mathbf{X}^* + \begin{bmatrix} -1, & -3 \\ 3, & 1 \end{bmatrix}$ and set $\mathbf{A} = \begin{bmatrix} 10, & 10 \\ 10, & 10 \end{bmatrix}$, $K = 10^9$, $\theta = 1 - \frac{1}{\sqrt{K}}$, $\beta = \sqrt{\theta}$, $\eta = \frac{1}{\sqrt{K}}$, $\varepsilon = 10^{-12}$, $p = q = 2$, $\mathbf{M}_0 = \mathbf{0}$, $\mathbf{L}_0 = \mathbf{0}$, $\mathbf{R}_0 = \mathbf{0}$ for Algorithm 1. We test $\lambda = \{10^{-1}, 10^{-2}, 10^{-3}, 10^{-4}, 10^{-5}, 0\}$ such that $\|\mathbf{X}^*\|_{op} = 8 < \frac{1}{\lambda}$ and $\|\mathbf{X}_1\|_{op} \le 9.2 < \frac{1}{\lambda}$. Figure 1 demonstrates that Algorithm 1 fails to converge to $\mathbf{X}^*$ when $\lambda = \{10^{-1}, 10^{-2}, 10^{-3}, 10^{-4}\}$. This indicates that, even for a simple convex problem, convergence to the minimum solution is not guaranteed if $\lambda$ exceeds a certain threshold, although it is unclear whether our upper bound is tight.

**Comparison with (Xie et al., 2026)**. While Xie et al. (2026) established convergence for adaptive optimizers under a nonconvex, adaptive smoothness framework, our work

differs in several key aspects. Firstly, we provide a unified treatment of two-sided and one-sided Shampoo, addressing the more complex and widely used former variant, while Xie et al. (2026) only studied the latter. Secondly, we incorporate practical components like momentum and decoupled weight decay which Xie et al. (2026) omits. Thirdly, we utilize a stochastic expectation-based assumption, which is weaker than the deterministic condition $-\Sigma_L \preceq \mathbf{G}\mathbf{G}^T - \nabla f(\mathbf{X})\nabla f(\mathbf{X})^T \preceq \Sigma_L$ used in (Xie et al., 2026). Finally, while both analyses achieve the $\mathcal{O}(\frac{C}{K^{1/4}})$ convergence rate, our constant $C$ in (7) is simpler and matches that in the optimal convergence rate of SGD under the standard Euclidean smoothness assumption.

## 3. Proof of the Theorem

We present our techniques to address the challenging two-sided preconditioning in Sections 3.1 and 3.2, which constitute the primary technical contribution of this paper relative to existing literature. Section 3.3 then outlines the proof sketch of Theorem 2.1.

### 3.1. Bounding the Nuclear Norm of Gradient by Schatten-$p$ Holder Inequality

In the analysis of AdamW, Li et al. (2025a) employ the following inequality to measure the expected $\ell_1$ norm summation of gradients

$$\left(\sum_{k=1}^{K}\mathbb{E}\left[\|\nabla f(\mathbf{x}_k)\|_1\right]\right)^2$$

$$\le \left(\sum_{k=1}^{K}\sum_{i=1}^{d}\mathbb{E}\left[\frac{|\nabla_i f(\mathbf{x}_k)|^2}{\sqrt{\widetilde{\mathbf{v}}_{k,i}}+\varepsilon}\right]\right)\left(\sum_{k=1}^{K}\sum_{i=1}^{d}\mathbb{E}\left[\sqrt{\widetilde{\mathbf{v}}_{k,i}}+\varepsilon\right]\right) \quad (9)$$

by Holder's inequality for some $\widetilde{\mathbf{v}}_k$ to approximate the second moment, where $\widetilde{\mathbf{v}}_{k,i}$ denotes the $i$-th element of vector $\widetilde{\mathbf{v}}$ at the $k$-th iteration. To handle the more complex matrix case with two-sided preconditioning, we instead utilize the Schatten-$p$ Holder inequality.

**Definition 3.1.** (Bhatia, 1997, (IV.31)) For any matrix $\mathbf{A} \in \mathbb{R}^{m \times n}$ with $r = \min\{m, n\}$, the Schatten-$p$ norm is defined via its singular values as

$$\|\mathbf{A}\|_{S_p} = \begin{cases} \left(\sum_{i=1}^{r}(\sigma_i(\mathbf{A}))^p\right)^{1/p}, & 1 \le p < \infty, \\ \sigma_1(\mathbf{A}), & p = \infty. \end{cases}$$

The following lemma extends Holder's inequality to the Schatten-$p$ norm.

**Lemma 3.2.** *(Bhatia, 1997, Corollary IV.2.6, Exercise IV.2.7,(IV.33)) Let $\mathbf{A}_i \in \mathbb{R}^{m \times m}$ and $p_i$ be positive real number $(i = 1, 2, \cdots, t)$ such that $\sum_{i=1}^{t}\frac{1}{p_i} = 1$. Then*

$$\left\|\Pi_{i=1}^{t}\mathbf{A}_i\right\|_{S_1} \le \Pi_{i=1}^{t}\|\mathbf{A}_i\|_{S_{p_i}}.$$

Note that we do not require $\mathbf{A}_i$ to be symmetric positive semidefinite. The above inequality also holds for non-square matrices, because we can always obtain a square matrix by appending zero columns to the right and zero rows to the bottom of the original matrix, and these appended zeros do not affect the Schatten-$p$ norm and matrix multiplication.

The following lemma generalizes (9) from vectors to matrices.

**Lemma 3.3.** *Let $\frac{1}{p} + \frac{1}{q} = 1$. Then for Algorithm 1, we have*

$$\sum_{k=1}^{K}\mathbb{E}\left[\|\nabla f(\mathbf{X}_k)\|_*\right] \le \left(\sum_{k=1}^{K}\mathbb{E}\left[\mathrm{tr}\left(\mathbf{L}_{k,\varepsilon}^{1/2}\right) + \mathrm{tr}\left(\mathbf{R}_{k,\varepsilon}^{1/2}\right)\right]\right)^{1/2}$$

$$\times \left(\sum_{k=1}^{K}\mathbb{E}\left[\left\|\mathbf{L}_{k,\varepsilon}^{-\frac{1}{4p}}\nabla f(\mathbf{X}_k)\mathbf{R}_{k,\varepsilon}^{-\frac{1}{4q}}\right\|_F^2\right]\right)^{1/2}.$$

*Proof.* From the definitions of the Schatten-$p$ norm and nuclear norm, we have

$$\sum_{k=1}^{K}\mathbb{E}\left[\|\nabla f(\mathbf{X}_k)\|_*\right] = \sum_{k=1}^{K}\mathbb{E}\left[\|\nabla f(\mathbf{X}_k)\|_{S_1}\right]$$

$$= \sum_{k=1}^{K}\mathbb{E}\left[\left\|\mathbf{L}_{k,\varepsilon}^{\frac{1}{4p}}\mathbf{L}_{k,\varepsilon}^{-\frac{1}{4p}}\nabla f(\mathbf{X}_k)\mathbf{R}_{k,\varepsilon}^{-\frac{1}{4q}}\mathbf{R}_{k,\varepsilon}^{\frac{1}{4q}}\right\|_{S_1}\right]$$

$$\overset{a}{\le} \sum_{k=1}^{K}\mathbb{E}\left[\left\|\mathbf{L}_{k,\varepsilon}^{\frac{1}{4p}}\right\|_{S_{2p}}\left\|\mathbf{L}_{k,\varepsilon}^{-\frac{1}{4p}}\nabla f(\mathbf{X}_k)\mathbf{R}_{k,\varepsilon}^{-\frac{1}{4q}}\right\|_{S_2}\left\|\mathbf{R}_{k,\varepsilon}^{\frac{1}{4q}}\right\|_{S_{2q}}\right]$$

$$\overset{b}{\le} \left(\sum_{k=1}^{K}\mathbb{E}\left[\left\|\mathbf{L}_{k,\varepsilon}^{\frac{1}{4p}}\right\|_{S_{2p}}^{2p}\right]\right)^{1/2p}\left(\sum_{k=1}^{K}\mathbb{E}\left[\left\|\mathbf{R}_{k,\varepsilon}^{\frac{1}{4q}}\right\|_{S_{2q}}^{2q}\right]\right)^{1/2q}$$

$$\times \left(\sum_{k=1}^{K}\mathbb{E}\left[\left\|\mathbf{L}_{k,\varepsilon}^{-\frac{1}{4p}}\nabla f(\mathbf{X}_k)\mathbf{R}_{k,\varepsilon}^{-\frac{1}{4q}}\right\|_{S_2}^2\right]\right)^{1/2},$$

where we use $\frac{1}{2p} + \frac{1}{2q} + \frac{1}{2} = 1$ and Lemma 3.2 of the Schatten-$p$ Holder inequality in $\overset{a}{\le}$, and Holder's inequality in $\overset{b}{\le}$.

Denote $\mathbf{A} = \mathbf{L}_{k,\varepsilon}^{-\frac{1}{4p}}\nabla f(\mathbf{X}_k)\mathbf{R}_{k,\varepsilon}^{-\frac{1}{4q}} \in \mathbb{R}^{m\times n}$ with $r = \min\{m, n\}$. Then from Definition 3.1 of the Schatten-$p$ norm, we have

$$\|\mathbf{A}\|_{S_2}^2 = \sum_{i=1}^{r}(\sigma_i(\mathbf{A}))^2 = \mathrm{tr}(\mathbf{A}^T\mathbf{A}) = \|\mathbf{A}\|_F^2.$$

When $p, q < +\infty$, from Definition 3.1 and the fact that $\mathbf{L}_{k,\varepsilon}$ is symmetric positive definite, we have

$$\left\|\mathbf{L}_{k,\varepsilon}^{\frac{1}{4p}}\right\|_{S_{2p}}^{2p} = \left(\sum_{i=1}^{m}\left(\sigma_i\left(\mathbf{L}_{k,\varepsilon}^{\frac{1}{4p}}\right)\right)^{2p}\right)^{\frac{1}{2p}\cdot 2p} \overset{c}{=} \mathrm{tr}\left(\mathbf{L}_{k,\varepsilon}^{1/2}\right),(10)$$

where we use $\sigma_i(\mathbf{B}^{\frac{1}{p}}) = (\sigma_i(\mathbf{B}))^{\frac{1}{p}}$ for any symmetric positive semidefinite matrix $\mathbf{B}$ and Fact B.2 in $\overset{c}{=}$. The derivation for $\mathbf{R}_{k,\varepsilon}$ follows a similar approach. Using $a^{\frac{1}{2p}}b^{\frac{1}{2q}} \le (a+b)^{\frac{1}{2p}}(a+b)^{\frac{1}{2q}} = (a+b)^{1/2}$ for positive $a, b$, we have the conclusion.

When $p = +\infty$ and $q = 1$, we have $\mathbf{L}_{k,\varepsilon}^{\frac{1}{4p}} = \mathbf{I}_m$, $\mathbf{L}_{k,\varepsilon}^{-\frac{1}{4p}} = \mathbf{I}_m$, and $\left\|\mathbf{L}_{k,\varepsilon}^{\frac{1}{4p}}\right\|_{S_{2p}} = 1$. So we have

$$\sum_{k=1}^{K}\mathbb{E}\left[\|\nabla f(\mathbf{X}_k)\|_*\right] \le \left(\sum_{k=1}^{K}\mathbb{E}\left[\mathrm{tr}\left(\mathbf{R}_{k,\varepsilon}^{1/2}\right)\right]\right)^{1/2}$$

$$\times \left(\sum_{k=1}^{K}\mathbb{E}\left[\left\|\mathbf{L}_{k,\varepsilon}^{-\frac{1}{4p}}\nabla f(\mathbf{X}_k)\mathbf{R}_{k,\varepsilon}^{-\frac{1}{4q}}\right\|_F^2\right]\right)^{1/2}.$$

The case when $q = +\infty$ and $p = 1$ is similar. $\square$

The essence of Lemma 3.3 lies in transforming $\mathbf{L}_{k,\varepsilon}^{-\frac{1}{4p}}$ and $\mathbf{R}_{k,\varepsilon}^{-\frac{1}{4q}}$ into the more tractable $\mathbf{L}_{k,\varepsilon}^{1/2}$ and $\mathbf{R}_{k,\varepsilon}^{1/2}$, respectively, which can be handled by the following lemma.

**Lemma 3.4.** *Suppose that Assumptions 2-3 hold. Let $\beta < 1$. Then for Algorithm 1, we have*

$$\sum_{k=1}^{K}\mathbb{E}_{\mathcal{F}_k}\left[\mathrm{tr}\left(\mathbf{L}_{k,\varepsilon}^{1/2}\right)\right]$$

$$\le K\,\mathrm{tr}\left(\Sigma_L^{1/2}\right) + Km\sqrt{\varepsilon} + \frac{2}{\sqrt{1-\beta}}\sum_{t=1}^{K}\mathbb{E}_{\mathcal{F}_{t-1}}[\|\nabla f(\mathbf{X}_t)\|_*],$$

$$\sum_{k=1}^{K}\mathbb{E}_{\mathcal{F}_k}\left[\mathrm{tr}\left(\mathbf{R}_{k,\varepsilon}^{1/2}\right)\right]$$

$$\le K\,\mathrm{tr}\left(\Sigma_R^{1/2}\right) + Kn\sqrt{\varepsilon} + \frac{2}{\sqrt{1-\beta}}\sum_{t=1}^{K}\mathbb{E}_{\mathcal{F}_{t-1}}[\|\nabla f(\mathbf{X}_t)\|_*].$$

*Proof.* From the recursion of $\mathbf{L}_{k-t}$, we have

$$\mathbb{E}_{\mathcal{F}_{k-t}}\left[\mathrm{tr}\left(\left(\beta^t\mathbf{L}_{k-t} + (1-\beta^t)\Sigma_L + \varepsilon\mathbf{I}_m\right)^{1/2}\right)\right]$$

$$= \mathbb{E}_{\mathcal{F}_{k-t}}\left[\mathrm{tr}\left(\left(\beta^{t+1}\mathbf{L}_{k-t-1} + \beta^t(1-\beta)\mathbf{G}_{k-t}\mathbf{G}_{k-t}^T + (1-\beta^t)\Sigma_L + \varepsilon\mathbf{I}_m\right)^{1/2}\right)\right]$$

$$= \mathbb{E}_{\mathcal{F}_{k-t-1}}\left[\mathbb{E}_{k-t}\left[\mathrm{tr}\left(\left(\beta^{t+1}\mathbf{L}_{k-t-1} + \beta^t(1-\beta)\mathbf{G}_{k-t}\mathbf{G}_{k-t}^T + (1-\beta^t)\Sigma_L + \varepsilon\mathbf{I}_m\right)^{1/2}\right)\big|\mathcal{F}_{k-t-1}\right]\right]$$

$$\overset{a}{\le} \mathbb{E}_{\mathcal{F}_{k-t-1}}\left[\mathrm{tr}\left(\left(\beta^{t+1}\mathbf{L}_{k-t-1} + (1-\beta^t)\Sigma_L + \varepsilon\mathbf{I}_m + \beta^t(1-\beta)\mathbb{E}_{k-t}\left[\mathbf{G}_{k-t}\mathbf{G}_{k-t}^T|\mathcal{F}_{k-t-1}\right]\right)^{1/2}\right)\right]$$

$$\overset{b}{\le} \mathbb{E}_{\mathcal{F}_{k-t-1}}\left[\mathrm{tr}\left(\left(\beta^{t+1}\mathbf{L}_{k-t-1} + (1-\beta^t)\Sigma_L + \varepsilon\mathbf{I}_m + \beta^t(1-\beta)\nabla f(\mathbf{X}_{k-t})\nabla f(\mathbf{X}_{k-t})^T + \beta^t(1-\beta)\Sigma_L\right)^{1/2}\right)\right]$$

$$=\mathbb{E}_{\mathcal{F}_{k-t-1}}\left[\operatorname{tr}\left((\beta^{t+1}\mathbf{L}_{k-t-1}+(1-\beta^{t+1})\Sigma_L+\varepsilon\mathbf{I}_m\right.\right.$$
$$\left.\left.+\beta^t(1-\beta)\nabla f(\mathbf{X}_{k-t})\nabla f(\mathbf{X}_{k-t})^T\right)^{1/2}\right)\right]$$
$$\overset{c}{\leq}\mathbb{E}_{\mathcal{F}_{k-t-1}}\left[\operatorname{tr}\left((\beta^{t+1}\mathbf{L}_{k-t-1}+(1-\beta^{t+1})\Sigma_L+\varepsilon\mathbf{I}_m)^{1/2}\right)\right.$$
$$\left.+\sqrt{\beta^t(1-\beta)}\operatorname{tr}\left((\nabla f(\mathbf{X}_{k-t})\nabla f(\mathbf{X}_{k-t})^T)^{1/2}\right)\right]$$
$$\overset{d}{=}\mathbb{E}_{\mathcal{F}_{k-t-1}}\left[\operatorname{tr}\left((\beta^{t+1}\mathbf{L}_{k-t-1}+(1-\beta^{t+1})\Sigma_L+\varepsilon\mathbf{I}_m)^{1/2}\right)\right.$$
$$\left.+\sqrt{\beta^t(1-\beta)}\,\|\nabla f(\mathbf{X}_{k-t})\|_*\right],$$

where we use the concavity of $\mathbf{X}^{1/2}$ presented in Lemma B.7 and $\operatorname{tr}(\mathbf{X}) \leq \operatorname{tr}(\mathbf{Y})$ if $\mathbf{X} \preceq \mathbf{Y}$ presented in Lemma B.1 in $\overset{a}{\leq}$, Assumptions 2-3 and the monotonicity of $\mathbf{X}^{1/2}$ presented in Lemma B.6 in $\overset{b}{\leq}$, the property $\operatorname{tr}\left((\mathbf{X}+\mathbf{Y})^{1/2}\right) \leq \operatorname{tr}(\mathbf{X}^{1/2}) + \operatorname{tr}(\mathbf{Y}^{1/2})$ for symmetric positive semidefinite matrices presented in Lemma B.4 in $\overset{c}{\leq}$, and Lemma B.5 in $\overset{d}{=}$. Applying the above inequality recursively for $t = 0, 1, 2, \cdots, k-1$, we have

$$\mathbb{E}_{\mathcal{F}_k}\left[\operatorname{tr}\left(\mathbf{L}_{k,\varepsilon}^{1/2}\right)\right]$$
$$\leq \operatorname{tr}\left((\beta^k\mathbf{L}_0+(1-\beta^k)\Sigma_L+\varepsilon\mathbf{I}_m)^{1/2}\right)$$
$$+\sqrt{1-\beta}\sum_{t=0}^{k-1}\sqrt{\beta^t}\mathbb{E}_{\mathcal{F}_{k-t-1}}\left[\|\nabla f(\mathbf{X}_{k-t})\|_*\right]$$
$$\overset{e}{\leq}\operatorname{tr}\left(\Sigma_L^{1/2}+\sqrt{\varepsilon}\mathbf{I}_m\right)+\sqrt{1-\beta}\sum_{t=1}^{k}\sqrt{\beta^{k-t}}\mathbb{E}_{\mathcal{F}_{t-1}}[\|\nabla f(\mathbf{X}_t)\|_*]$$
$$=\operatorname{tr}\left(\Sigma_L^{1/2}\right)+m\sqrt{\varepsilon}+\sqrt{1-\beta}\sum_{t=1}^{k}\sqrt{\beta^{k-t}}\mathbb{E}_{\mathcal{F}_{t-1}}[\|\nabla f(\mathbf{X}_t)\|_*],$$

where we use $\mathbf{L}_0 = \mathbf{0}$, monotonicity of $\mathbf{X}^{1/2}$, and $\operatorname{tr}\left((\mathbf{X}+\mathbf{Y})^{1/2}\right) \leq \operatorname{tr}(\mathbf{X}^{1/2})+\operatorname{tr}(\mathbf{Y}^{1/2})$ in $\overset{e}{\leq}$. Summing over $k = 1, 2, \cdots, K$, we have

$$\sum_{k=1}^{K}\mathbb{E}_{\mathcal{F}_k}\left[\operatorname{tr}\left(\mathbf{L}_{k,\varepsilon}^{1/2}\right)\right] \leq K\operatorname{tr}\left(\Sigma_L^{1/2}\right)+Km\sqrt{\varepsilon}$$
$$+\underbrace{\sqrt{1-\beta}\sum_{k=1}^{K}\sum_{t=1}^{k}\sqrt{\beta^{k-t}}\mathbb{E}_{\mathcal{F}_{t-1}}\left[\|\nabla f(\mathbf{X}_t)\|_*\right]}_{\text{term (a)}}$$

and term (a) can be addressed as follows

$$\text{term (a)}=\sqrt{1-\beta}\sum_{t=1}^{K}\sum_{k=t}^{K}\sqrt{\beta^{k-t}}\mathbb{E}_{\mathcal{F}_{t-1}}\left[\|\nabla f(\mathbf{X}_t)\|_*\right]$$
$$\leq \frac{\sqrt{1-\beta}}{1-\sqrt{\beta}}\sum_{t=1}^{K}\mathbb{E}_{\mathcal{F}_{t-1}}\left[\|\nabla f(\mathbf{X}_t)\|_*\right]$$
$$\leq \frac{2}{\sqrt{1-\beta}}\sum_{t=1}^{K}\mathbb{E}_{\mathcal{F}_{t-1}}\left[\|\nabla f(\mathbf{X}_t)\|_*\right].$$

Similarly, we also have the inequality for $\mathbf{R}_{k,\varepsilon}^{1/2}$. $\qquad\square$

Combining Lemmas 3.3 and 3.4, we finally have

$$\sum_{k=1}^{K}\mathbb{E}[\|\nabla f(\mathbf{X}_k)\|_*]\leq\left(KC+\frac{4}{\sqrt{1-\beta}}\sum_{t=1}^{K}\mathbb{E}[\|\nabla f(\mathbf{X}_t)\|_*]\right)^{1/2}$$
$$\times\underbrace{\left(\sum_{k=1}^{K}\mathbb{E}\left[\left\|\mathbf{L}_{k,\varepsilon}^{-\frac{1}{4p}}\nabla f(\mathbf{X}_k)\mathbf{R}_{k,\varepsilon}^{-\frac{1}{4q}}\right\|_F^2\right]\right)^{1/2}}_{\text{term (b)}}, \quad (11)$$

where $C = \operatorname{tr}\left(\Sigma_L^{1/2}\right) + \operatorname{tr}\left(\Sigma_R^{1/2}\right) + (m+n)\sqrt{\varepsilon}$. So we only need to bound term (b).

### 3.2. Bounding the Spectral Norm of Update by Matrix Cauchy-Schwarz Inequality

In the analysis of AdamW (Li et al., 2025a), we can bound the update $\frac{|\mathbf{m}_{k,i}|}{\sqrt{\mathbf{v}_{k,i}}}$ coordinately, where $\mathbf{m}$ and $\mathbf{v}$ are the first and second moments, respectively. However, the matrix case is not as simple, especially with two-sided preconditioning. To address this challenge, we use the following matrix Cauchy-Schwarz inequality.

**Lemma 3.5.** *(Bhatia, 1997, Corollary IX.5.3) For* $\mathbf{M} \in \mathbb{R}^{m \times n}$ *and symmetric positive definite matrices* $\mathbf{L} \in \mathbb{R}^{m \times m}$ *and* $\mathbf{R} \in \mathbb{R}^{n \times n}$, $0 \leq \alpha \leq 1$, *we have*

$$\|\mathbf{L}^{\alpha}\mathbf{M}\mathbf{R}^{1-\alpha}\|_{op} \leq \|\mathbf{L}\mathbf{M}\|_{op}^{\alpha}\|\mathbf{M}\mathbf{R}\|_{op}^{1-\alpha}.$$

Based on the above lemma, we can bound the update in Algorithm 1 measured by spectral norm. Note that Lemma 3.6 still holds even when $\mathbf{L}_{k,\varepsilon}$ and $\mathbf{R}_{k,\varepsilon}$ are ill-conditioned.

**Lemma 3.6.** *Let* $\theta \leq \beta \leq \sqrt{\theta} < 1$ *and* $\frac{1}{p} + \frac{1}{q} = 1$. *Then for Algorithm 1, we have*

$$\left\|\mathbf{L}_{k,\varepsilon}^{-\frac{1}{2p}}\mathbf{M}_k\mathbf{R}_{k,\varepsilon}^{-\frac{1}{2q}}\right\|_{op} \leq 2.$$

*Proof.* From Lemma 3.5 and $\frac{1}{p} + \frac{1}{q} = 1$, we have

$$\left\|\mathbf{L}_{k,\varepsilon}^{-\frac{1}{2p}}\mathbf{M}_k\mathbf{R}_{k,\varepsilon}^{-\frac{1}{2q}}\right\|_{op} \leq \left\|\mathbf{L}_{k,\varepsilon}^{-\frac{1}{2}}\mathbf{M}_k\right\|_{op}^{\frac{1}{p}}\left\|\mathbf{M}_k\mathbf{R}_{k,\varepsilon}^{-\frac{1}{2}}\right\|_{op}^{\frac{1}{q}}.$$

So we only need to prove

$$\left\|\mathbf{L}_{k,\varepsilon}^{-\frac{1}{2}}\mathbf{M}_k\right\|_{op} \leq 2 \quad \text{and} \quad \left\|\mathbf{M}_k\mathbf{R}_{k,\varepsilon}^{-\frac{1}{2}}\right\|_{op} \leq 2.$$

The two inequalities are analogous, so we prove only the first. Since $\left\|\mathbf{L}_{k,\varepsilon}^{-\frac{1}{2}}\mathbf{M}_k\right\|_{op}^2 = \left\|\mathbf{L}_{k,\varepsilon}^{-\frac{1}{2}}\mathbf{M}_k\mathbf{M}_k^T\mathbf{L}_{k,\varepsilon}^{-\frac{1}{2}}\right\|_{op}$, we only need to prove

$$\mathbf{L}_{k,\varepsilon}^{-\frac{1}{2}}\mathbf{M}_k\mathbf{M}_k^T\mathbf{L}_{k,\varepsilon}^{-\frac{1}{2}} \preceq 4\mathbf{I}_m.$$

Since $\mathbf{L}_{k,\varepsilon}$ is invertible, the above inequality is equivalent to

$$\mathbf{M}_k\mathbf{M}_k^T \preceq 4\mathbf{L}_{k,\varepsilon} \text{ and } \mathbf{y}^T\mathbf{M}_k\mathbf{M}_k^T\mathbf{y} \leq 4\mathbf{y}^T\mathbf{L}_{k,\varepsilon}\mathbf{y}, \forall \mathbf{y} \in \mathbb{R}^m.$$

From the recursions of $\mathbf{M}_k$ and $\mathbf{L}_k$, we have

$$\mathbf{y}^T\mathbf{M}_k = (1-\theta)\sum_{t=1}^{k}\theta^{k-t}\mathbf{y}^T\mathbf{G}_t,$$

$$\mathbf{y}^T\mathbf{L}_k\mathbf{y} = (1-\beta)\sum_{t=1}^{k}\beta^{k-t}\mathbf{y}^T\mathbf{G}_t\mathbf{G}_t^T\mathbf{y}$$

$$= (1-\beta)\sum_{t=1}^{k}\beta^{k-t}\left\|\mathbf{y}^T\mathbf{G}_t\right\|^2.$$

From Holder's inequality, we have

$$\mathbf{y}^T\mathbf{M}_k\mathbf{M}_k^T\mathbf{y} = (1-\theta)^2\left\|\sum_{t=1}^{k}\theta^{k-t}\mathbf{y}^T\mathbf{G}_t\right\|^2$$

$$\leq (1-\theta)^2\left(\sum_{t=1}^{k}\theta^{k-t}\left\|\mathbf{y}^T\mathbf{G}_t\right\|\right)^2$$

$$\leq (1-\theta)^2\left(\sum_{t=1}^{k}\beta^{k-t}\left\|\mathbf{y}^T\mathbf{G}_t\right\|^2\right)\left(\sum_{t=1}^{k}\left(\frac{\theta^2}{\beta}\right)^{k-t}\right)$$

$$= \frac{(1-\theta)^2}{1-\beta}\mathbf{y}^T\mathbf{L}_k\mathbf{y}\left(\sum_{t=1}^{k}\left(\frac{\theta^2}{\beta}\right)^{k-t}\right)$$

$$\leq \frac{(1-\theta)^2}{1-\beta}\frac{1}{1-\frac{\theta^2}{\beta}}\mathbf{y}^T\mathbf{L}_k\mathbf{y} \overset{a}{\leq} \frac{(1-\theta)^2}{(1-\beta)^2}\mathbf{y}^T\mathbf{L}_k\mathbf{y}$$

$$\overset{b}{\leq} \frac{(1-\sqrt{\theta})^2(1+\sqrt{\theta})^2}{(1-\sqrt{\theta})^2}\mathbf{y}^T\mathbf{L}_k\mathbf{y} \leq 4\mathbf{y}^T\mathbf{L}_k\mathbf{y} \leq 4\mathbf{y}^T\mathbf{L}_{k,\varepsilon}\mathbf{y},$$

where we use $\theta \leq \beta$ in $\overset{a}{\leq}$ and $\beta \leq \sqrt{\theta}$ in $\overset{b}{\leq}$. $\qquad\square$

*Remark* 3.7. The requirement $\beta \leq \sqrt{\theta}$ excludes the common setting $(\theta, \beta) = (0.9, 0.999)$. In fact, we can relax this condition to $\beta \leq \theta^{1/128}$, noting that $0.9^{1/128} \geq 0.999$. In this case, the induction step in $\overset{b}{\leq}$ should be replaced by $\frac{(1-\theta)^2}{(1-\beta)^2} \leq \frac{(1-\theta^{1/2^7})^2\Pi_{r=1}^{7}(1+\theta^{1/2^r})^2}{(1-\theta^{1/2^7})^2} \leq 4^7$, at the cost of a larger constant term.

By leveraging Lemma 3.6 and the distinct properties of decoupled weight decay, we ultimately establish the following lemma.

**Lemma 3.8.** *Let* $\eta\lambda \leq \frac{\sqrt{\nu}}{2K^{5/4}}$, $\|\mathbf{X}_1\|_{op} \leq \frac{\sqrt{\nu}}{K^{1/4}\lambda}$, $\frac{\sqrt{\nu}}{K^{1/4}} \leq 1$, $\theta \leq \beta \leq \sqrt{\theta} < 1$, *and* $\frac{1}{p}+\frac{1}{q} = 1$ *for some constant* $\nu$. *Then for Algorithm 1, we have*

$$\lambda\|\mathbf{X}_k\|_{op} \leq \frac{3\sqrt{\nu}}{K^{1/4}}, \quad \forall k = 1, 2, \cdots, K. \tag{12}$$

### 3.3. Proof Sketch of Theorem 2.1

Building upon the supporting lemmas in Sections 3.1 and 3.2, we can prove Theorem 2.1 following the framework in (Li et al., 2025a). We briefly outline the proof sketch.

From the Lipschitz smoothness and the update of $\mathbf{X}_{k+1}$, we have

$$f(\mathbf{X}_{k+1})-f(\mathbf{X}_k) \leq \langle\nabla f(\mathbf{X}_k), \mathbf{X}_{k+1}-\mathbf{X}_k\rangle + \frac{L}{2}\|\mathbf{X}_{k+1}-\mathbf{X}_k\|_F^2$$

$$= -\eta\left\langle\nabla f(\mathbf{X}_k), \lambda\mathbf{X}_k + \mathbf{L}_{k,\varepsilon}^{-\frac{1}{2p}}\mathbf{M}_k\mathbf{R}_{k,\varepsilon}^{-\frac{1}{2q}}\right\rangle$$

$$+ \frac{L\eta^2}{2}\left\|\lambda\mathbf{X}_k + \mathbf{L}_{k,\varepsilon}^{-\frac{1}{2p}}\mathbf{M}_k\mathbf{R}_{k,\varepsilon}^{-\frac{1}{2q}}\right\|_F^2$$

$$= \underbrace{-\eta\left\langle\mathbf{L}_{k,\varepsilon}^{-\frac{1}{4p}}\nabla f(\mathbf{X}_k)\mathbf{R}_{k,\varepsilon}^{-\frac{1}{4q}}, \lambda\mathbf{L}_{k,\varepsilon}^{\frac{1}{4p}}\mathbf{X}_k\mathbf{R}_{k,\varepsilon}^{\frac{1}{4q}}+\mathbf{L}_{k,\varepsilon}^{-\frac{1}{4p}}\mathbf{M}_k\mathbf{R}_{k,\varepsilon}^{-\frac{1}{4q}}\right\rangle}_{\text{term (c)}}$$

$$+ \underbrace{\frac{L\eta^2}{2}\left\|\mathbf{L}_{k,\varepsilon}^{-\frac{1}{4p}}\left(\lambda\mathbf{L}_{k,\varepsilon}^{\frac{1}{4p}}\mathbf{X}_k\mathbf{R}_{k,\varepsilon}^{\frac{1}{4q}}+\mathbf{L}_{k,\varepsilon}^{-\frac{1}{4p}}\mathbf{M}_k\mathbf{R}_{k,\varepsilon}^{-\frac{1}{4q}}\right)\mathbf{R}_{k,\varepsilon}^{-\frac{1}{4q}}\right\|_F^2}_{\text{term (d)}}.$$

Decompose term (c) into

$$-\frac{\eta}{2}\left\|\mathbf{L}_{k,\varepsilon}^{-\frac{1}{4p}}\nabla f(\mathbf{X}_k)\mathbf{R}_{k,\varepsilon}^{-\frac{1}{4q}}\right\|_F^2$$

$$-\frac{\eta}{2}\left\|\lambda\mathbf{L}_{k,\varepsilon}^{\frac{1}{4p}}\mathbf{X}_k\mathbf{R}_{k,\varepsilon}^{\frac{1}{4q}}+\mathbf{L}_{k,\varepsilon}^{-\frac{1}{4p}}\mathbf{M}_k\mathbf{R}_{k,\varepsilon}^{-\frac{1}{4q}}\right\|_F^2$$

$$+\underbrace{\frac{\eta}{2}\left\|\mathbf{L}_{k,\varepsilon}^{-\frac{1}{4p}}\left(\nabla f(\mathbf{X}_k)-\mathbf{M}_k\right)\mathbf{R}_{k,\varepsilon}^{-\frac{1}{4q}}-\lambda\mathbf{L}_{k,\varepsilon}^{\frac{1}{4p}}\mathbf{X}_k\mathbf{R}_{k,\varepsilon}^{\frac{1}{4q}}\right\|_F^2}_{\text{term (e)}}$$

and relax term (e) to

$$\underbrace{\eta\left\|\mathbf{L}_{k,\varepsilon}^{-\frac{1}{4p}}(\nabla f(\mathbf{X}_k)-\mathbf{M}_k)\mathbf{R}_{k,\varepsilon}^{-\frac{1}{4q}}\right\|_F^2}_{\text{term (f)}}+\underbrace{\eta\lambda^2\left\|\mathbf{L}_{k,\varepsilon}^{\frac{1}{4p}}\mathbf{X}_k\mathbf{R}_{k,\varepsilon}^{\frac{1}{4q}}\right\|_F^2}_{\text{term (g)}}.$$

From condition (4), we can relax terms (f) and (d) as follow

$$\text{term (f)} \leq \frac{\eta}{\sqrt{\hat{\varepsilon}}}\|\nabla f(\mathbf{X}_k)-\mathbf{M}_k\|_F^2,$$

$$\text{term (d)} \leq \frac{L\eta^2}{2\sqrt{\hat{\varepsilon}}}\left\|\lambda\mathbf{L}_{k,\varepsilon}^{\frac{1}{4p}}\mathbf{X}_k\mathbf{R}_{k,\varepsilon}^{\frac{1}{4q}}+\mathbf{L}_{k,\varepsilon}^{-\frac{1}{4p}}\mathbf{M}_k\mathbf{R}_{k,\varepsilon}^{-\frac{1}{4q}}\right\|_F^2.$$

The handling of term (g) requires a high degree of skill and we present the details here.

$$\begin{aligned}
\text{(g)} &= \text{tr}\left(\mathbf{L}_{k,\varepsilon}^{\frac{1}{4p}}\mathbf{X}_k\mathbf{R}_{k,\varepsilon}^{\frac{1}{2q}}\mathbf{X}_k^T\mathbf{L}_{k,\varepsilon}^{\frac{1}{4p}}\right) \overset{a}{=} \left\|\mathbf{L}_{k,\varepsilon}^{\frac{1}{4p}}\mathbf{X}_k\mathbf{R}_{k,\varepsilon}^{\frac{1}{2q}}\mathbf{X}_k^T\mathbf{L}_{k,\varepsilon}^{\frac{1}{4p}}\right\|_{S_1}\\
&\overset{b}{\leq} \left\|\mathbf{L}_{k,\varepsilon}^{\frac{1}{4p}}\right\|_{S_{2p}}\left\|\mathbf{X}_k\mathbf{R}_{k,\varepsilon}^{\frac{1}{2q}}\mathbf{X}_k^T\right\|_{S_q}\left\|\mathbf{L}_{k,\varepsilon}^{\frac{1}{4p}}\right\|_{S_{2p}}\\
&\overset{c}{=} \left(\text{tr}\left(\mathbf{L}_{k,\varepsilon}^{1/2}\right)\right)^{\frac{1}{p}}\underbrace{\left\|\mathbf{X}_k\mathbf{R}_{k,\varepsilon}^{\frac{1}{2q}}\mathbf{X}_k^T\right\|_{S_q}}_{\text{term (h)}},
\end{aligned} \tag{13}$$

where we use Definition 3.1 of the Schatten-$p$ norm, Fact B.2, and the fact that $\mathbf{L}_{k,\varepsilon}^{\frac{1}{4p}}\mathbf{X}_k\mathbf{R}_{k,\varepsilon}^{\frac{2}{2q}}\mathbf{X}_k^T\mathbf{L}_{k,\varepsilon}^{\frac{1}{4p}}$ is symmetric positive semidefinite in $\overset{a}{=}$, $\frac{1}{p}+\frac{1}{q}=1$ and Lemma 3.2 of the Schatten-$p$ Holder inequality in $\overset{b}{\leq}$, and (10) in $\overset{c}{=}$. For term (h), denoting $r$ to be the rank of $\mathbf{X}_k\mathbf{R}_{k,\varepsilon}^{\frac{1}{2q}}\mathbf{X}_k^T$, we have

term (h)

$$=\left(\sum_{i=1}^{r}\sigma_i\left(\mathbf{X}_k\mathbf{R}_{k,\varepsilon}^{\frac{1}{2q}}\mathbf{X}_k^T\right)^q\right)^{\frac{1}{q}}\overset{d}{\leq}\left(\sum_{i=1}^{n}\left(\|\mathbf{X}_k\|_{op}^2\sigma_i\left(\mathbf{R}_{k,\varepsilon}^{\frac{1}{2q}}\right)\right)^q\right)^{\frac{1}{q}}$$

$$\overset{e}{=}\left(\|\mathbf{X}_k\|_{op}^{2q}\sum_{i=1}^{n}\sigma_i\left(\mathbf{R}_{k,\varepsilon}^{1/2}\right)\right)^{\frac{1}{q}}\overset{f}{=}\|\mathbf{X}_k\|_{op}^2\left(\text{tr}\left(\mathbf{R}_{k,\varepsilon}^{1/2}\right)\right)^{\frac{1}{q}},$$

where we use $r \leq \min\{m,n\}$ and the properties $\sigma_i(\mathbf{AB}) \leq \sigma_i(\mathbf{A})\|\mathbf{B}\|_{op}$ and $\sigma_i(\mathbf{AB}) \leq \|\mathbf{A}\|_{op}\sigma_i(\mathbf{B})$ of singular values in $\overset{d}{\leq}$, $\sigma_i(\mathbf{B}^{\frac{1}{q}}) = (\sigma_i(\mathbf{B}))^{\frac{1}{q}}$ for any symmetric positive semidefinite matrix $\mathbf{B}$ in $\overset{e}{=}$, and Fact B.2 in $\overset{f}{=}$. Plugging into (13), we have

$$\text{term (g)} \leq \|\mathbf{X}_k\|_{op}^2\left(\text{tr}\left(\mathbf{L}_{k,\varepsilon}^{1/2}\right)\right)^{\frac{1}{p}}\left(\text{tr}\left(\mathbf{R}_{k,\varepsilon}^{1/2}\right)\right)^{\frac{1}{q}}$$

$$\overset{g}{\leq}\|\mathbf{X}_k\|_{op}^2\left(\text{tr}\left(\mathbf{L}_{k,\varepsilon}^{1/2}\right)+\text{tr}\left(\mathbf{R}_{k,\varepsilon}^{1/2}\right)\right)$$

$$\overset{h}{\leq}\frac{9\nu}{\lambda^2 K^{1/2}}\left(\text{tr}\left(\mathbf{L}_{k,\varepsilon}^{1/2}\right)+\text{tr}\left(\mathbf{R}_{k,\varepsilon}^{1/2}\right)\right),$$

where we use $a^{\frac{1}{p}}b^{\frac{1}{q}} \leq (a+b)^{\frac{1}{p}+\frac{1}{q}} = a+b$ for positive $a,b$ in $\overset{g}{\leq}$, and Lemma 3.8 in $\overset{h}{\leq}$. The above derivation for term (g) is conducted for the case of $p,q < +\infty$. The conclusion remains valid when either $p$ or $q$ is infinite.

Combining the above results and letting $\eta \leq \frac{\sqrt{\hat{\varepsilon}}}{2L}$, we have

$$f(\mathbf{X}_{k+1}) - f(\mathbf{X}_k) \leq -\frac{\eta}{2}\left\|\mathbf{L}_{k,\varepsilon}^{-\frac{1}{4p}}\nabla f(\mathbf{X}_k)\mathbf{R}_{k,\varepsilon}^{-\frac{1}{4q}}\right\|_F^2$$

$$-\frac{\eta}{4}\left\|\lambda\mathbf{L}_{k,\varepsilon}^{\frac{1}{4p}}\mathbf{X}_k\mathbf{R}_{k,\varepsilon}^{\frac{1}{4q}}+\mathbf{L}_{k,\varepsilon}^{-\frac{1}{4p}}\mathbf{M}_k\mathbf{R}_{k,\varepsilon}^{-\frac{1}{4q}}\right\|_F^2 \qquad (14)$$

$$+\frac{\eta}{\sqrt{\hat{\varepsilon}}}\|\nabla f(\mathbf{X}_k)-\mathbf{M}_k\|_F^2+\frac{9\eta\nu}{K^{1/2}}\left(\text{tr}\left(\mathbf{L}_{k,\varepsilon}^{1/2}\right)+\text{tr}\left(\mathbf{R}_{k,\varepsilon}^{1/2}\right)\right).$$

Employing standard techniques in the analysis of momentum SGD, we can build a recursion (Lemma D.2) as follows

$$\mathbb{E}_{\mathcal{F}_k}\left[\|\nabla f(\mathbf{X}_k)-\mathbf{M}_k\|_F^2\right]$$

$$\leq\mathbb{E}_{\mathcal{F}_{k-1}}\left[\theta\|\nabla f(\mathbf{X}_{k-1})-\mathbf{M}_{k-1}\|_F^2+(1-\theta)^2\sigma^2\right.$$

$$\left.+\frac{L^2\eta^2}{(1-\theta)\sqrt{\hat{\varepsilon}}}\left\|\lambda\mathbf{L}_{k-1,\varepsilon}^{\frac{1}{4p}}\mathbf{X}_{k-1}\mathbf{R}_{k-1,\varepsilon}^{\frac{1}{4q}}+\mathbf{L}_{k-1,\varepsilon}^{-\frac{1}{4p}}\mathbf{M}_{k-1}\mathbf{R}_{k-1,\varepsilon}^{-\frac{1}{4q}}\right\|_F^2\right].$$

(15)

Multiplying both sides of (15) by $\frac{\eta}{\sqrt{\hat{\varepsilon}}(1-\theta)}$, adding it to (14),

letting $\eta^2 \leq \frac{\hat{\varepsilon}(1-\theta)^2}{4L^2}$, and arranging the terms, we have

$$\phi_{k+1} \leq \phi_k - \frac{\eta}{2}\mathbb{E}_{\mathcal{F}_k}\left[\left\|\mathbf{L}_{k,\varepsilon}^{-\frac{1}{4p}}\nabla f(\mathbf{X}_k)\mathbf{R}_{k,\varepsilon}^{-\frac{1}{4q}}\right\|_F^2\right]$$

$$+\frac{9\eta\nu}{K^{1/2}}\mathbb{E}_{\mathcal{F}_k}\left[\text{tr}\left(\mathbf{L}_{k,\varepsilon}^{1/2}\right)+\text{tr}\left(\mathbf{R}_{k,\varepsilon}^{1/2}\right)\right]+\frac{\eta(1-\theta)}{\sqrt{\hat{\varepsilon}}}\sigma^2,$$

(16)

where $\phi_{k+1}=\mathbb{E}_{\mathcal{F}_k}\left[f(\mathbf{X}_{k+1})-f^*+\frac{\eta\theta}{\sqrt{\hat{\varepsilon}}(1-\theta)}\|\nabla f(\mathbf{X}_k)-\mathbf{M}_k\|_F^2\right.$ $\left.+\frac{\eta}{4}\left\|\lambda\mathbf{L}_{k,\varepsilon}^{\frac{1}{4p}}\mathbf{X}_k\mathbf{R}_{k,\varepsilon}^{\frac{1}{4q}}+\mathbf{L}_{k,\varepsilon}^{-\frac{1}{4p}}\mathbf{M}_k\mathbf{R}_{k,\varepsilon}^{-\frac{1}{4q}}\right\|_F^2\right]$. Summing (16) over $k = 1, 2, \cdots, K$ and setting the parameters properly, we have

$$\sum_{k=1}^{K}\mathbb{E}_{\mathcal{F}_k}\left[\left\|\mathbf{L}_{k,\varepsilon}^{-\frac{1}{4p}}\nabla f(\mathbf{X}_k)\mathbf{R}_{k,\varepsilon}^{-\frac{1}{4q}}\right\|_F^2\right]$$

$$\leq\frac{18\nu}{K^{1/2}}\sum_{k=1}^{K}\mathbb{E}_{\mathcal{F}_k}\left[\text{tr}\left(\mathbf{L}_{k,\varepsilon}^{1/2}\right)+\text{tr}\left(\mathbf{R}_{k,\varepsilon}^{1/2}\right)\right]+C',$$

where $C' = 10\sqrt{\frac{K\hat{\sigma}^2 L(f(\mathbf{X}_1)-f^*)}{\hat{\varepsilon}}}$. Plugging into (11) and using Lemma 3.4, we have

$$\sum_{k=1}^{K}\mathbb{E}[\|\nabla f(\mathbf{X}_k)\|_*]\leq\left(KC+\frac{4}{\sqrt{1-\beta}}\sum_{t=1}^{K}\mathbb{E}[\|\nabla f(\mathbf{X}_t)\|_*]\right)^{\frac{1}{2}}\times$$

$$\left(\frac{18\nu}{K^{1/2}}\frac{4}{\sqrt{1-\beta}}\sum_{t=1}^{K}\mathbb{E}[\|\nabla f(\mathbf{X}_t)\|_*]+18\nu K^{1/2}C+C'\right)^{\frac{1}{2}},$$

where $C$ is defined at the end of Section 3.1. Under proper parameter settings such that $\frac{288\nu}{K^{1/2}(1-\beta)} \leq \frac{1}{2}$, we finally have

$$\left(\sum_{k=1}^{K}\mathbb{E}\left[\|\nabla f(\mathbf{X}_k)\|_*\right]\right)^2 \leq A\sum_{k=1}^{K}\mathbb{E}\left[\|\nabla f(\mathbf{X}_k)\|_*\right] + B$$

for some constants $A$ and $B$. Solving this inequality, we have the conclusion.

## 4. Conclusion

This paper studies the convergence properties of AdamW-style Shampoo with both one-sided and two-sided preconditioning. We establish the convergence rate $\frac{1}{K}\sum_{k=1}^{K}\mathbb{E}\left[\|\nabla f(\mathbf{X}_k)\|_*\right] \leq \mathcal{O}(\frac{\sqrt{m+n}C}{K^{1/4}})$ measured by nuclear norm. It can be considered to be analogous to the optimal $\frac{1}{K}\sum_{k=1}^{K}\mathbb{E}\left[\|\nabla f(\mathbf{X}_k)\|_F\right] \leq \mathcal{O}(\frac{C}{K^{1/4}})$ convergence rate of SGD in the ideal case of $\|\nabla f(\mathbf{X})\|_* = \Theta(\sqrt{\min\{m,n\}})\|\nabla f(\mathbf{X})\|_F$ and balanced $m$ and $n$.

## Acknowledgements

H. Li was supported by the NSF China (No. 62476142) and Z. Lin was supported by the NSF China (No. 62276004).

## Impact Statement

This study is primarily concerned with theoretical analysis and it does not yield direct negative societal impacts.

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

# A. Relationship between Shampoo and the Preconditioned Non-Euclidean Steepest Descent

Non-Euclidean steepest descent explores the direction of steepest loss decrease under different norms with the following update (Carlson et al., 2015b;a; 2016)

$$\mathbf{x}_{k+1} = \operatorname*{argmin}_{\mathbf{x}} \langle \mathbf{g}_k, \mathbf{x} - \mathbf{x}_k \rangle + \frac{1}{2\gamma} \|\mathbf{x} - \mathbf{x}_k\|_{\text{gen}}^2, \tag{17}$$

where $\|\cdot\|_{\text{gen}}$ denotes a general norm. The trust-region counterpart of (17) is given as follows (Bernstein & Newhouse, 2024; Crawshaw et al., 2025; Veprikov et al., 2025)

$$\mathbf{x}_{k+1} = \operatorname*{argmin}_{\|\mathbf{x} - \mathbf{x}_k\|_{\text{gen}} \leq \eta} \langle \mathbf{g}_k, \mathbf{x} - \mathbf{x}_k \rangle = \mathbf{x}_k + \eta \text{LMO}_{\|\cdot\|_{\text{gen}}}(\mathbf{g}_k), \tag{18}$$

where the linear minimization oracle (LMO) is defined as $\text{LMO}_{\|\cdot\|_{\text{gen}}}(\mathbf{g}) = \operatorname{argmin}_{\|\mathbf{u}\|_{\text{gen}}=1} \langle \mathbf{g}, \mathbf{u} \rangle$ (Pethick et al., 2025). When the norm is chosen as the $\ell_2$ norm, the $\ell_\infty$ norm, or the spectral norm, (18) reduces to normalized gradient descent (Hazan et al., 2015), SignSGD (Bernstein et al., 2018), and Muon (Jordan et al., 2024), respectively.

Equipping (17) with preconditioning yields the update (Carlson et al., 2015b; Veprikov et al., 2025)

$$\mathbf{x}_{k+1} = \operatorname*{argmin}_{\mathbf{x}} \langle \mathbf{g}_k, \mathbf{x} - \mathbf{x}_k \rangle + \frac{1}{2\gamma} \|\mathbf{x} - \mathbf{x}_k\|_{\mathbf{D}_k}^2, \tag{19}$$

where $\mathbf{D}_k$ is the preconditioner. It is well-known that Shampoo is a special case of (19) when the norm is the Euclidean $\ell_2$ norm and we set $\mathbf{x}_k = \text{vec}(\mathbf{X}_k)$, $\mathbf{g}_k = \text{vec}(\mathbf{G}_k)$, and $\mathbf{D}_k = (\mathbf{R}_k \otimes \mathbf{L}_k)^{1/4}$. Here, $\otimes$ denotes the Kronecker product and we utilize the identity $\text{vec}(\mathbf{L}^{-1/4}\mathbf{G}\mathbf{R}^{-1/4}) = (\mathbf{R} \otimes \mathbf{L})^{-1/4} \text{vec}(\mathbf{G})$. Shampoo can also be expressed directly in matrix form of (19) as

$$\mathbf{X}_{k+1} = \operatorname*{argmin}_{\mathbf{X}} \langle \mathbf{G}_k, \mathbf{X} - \mathbf{X}_k \rangle + \frac{1}{2\gamma} \text{tr}\left( (\mathbf{X} - \mathbf{X}_k)^T \mathbf{L}_k^{1/4} (\mathbf{X} - \mathbf{X}_k) \mathbf{R}_k^{1/4} \right),$$

where we use the identity $\text{tr}(\mathbf{X}^T \mathbf{L}^{1/4} \mathbf{X} \mathbf{R}^{1/4}) = \text{vec}(\mathbf{X})^T \text{vec}(\mathbf{L}^{1/4} \mathbf{X} \mathbf{R}^{1/4}) = \text{vec}(\mathbf{X})^T (\mathbf{R} \otimes \mathbf{L})^{1/4} \text{vec}(\mathbf{X})$.

From a theoretical standpoint, although convergence guarantees for non-Euclidean steepest descent have been established (Carlson et al., 2015b; Pethick et al., 2025), a unified convergence analysis for the preconditioned setting remains challenging due to the complex and dynamic nature of the preconditioners. Gratton & Toint (2026)[2] provided a unified convergence theory for adaptive first-order methods, including AdaNorm, full and diagonal AdaGrad, Shampoo, and AdaGo (Zhang et al., 2025). However, at the algorithmic level, Gratton & Toint (2026) only considered the AdaGrad-style update of the preconditioners in Shampoo—rather than the more practical RMSProp-style—and they omitted decoupled weight decay. At the theoretical level, they relied on stronger assumptions and obtained an $\mathcal{O}(d/\sqrt{K})$ convergence rate in the best case, rather than the more common $\mathcal{O}(1/K^{1/4})$ rate in nonconvex stochastic optimization. Moreover, they measured convergence for Shampoo via the Frobenius norm (which is considerably smaller than the nuclear norm used in our paper), and their rate exhibits an explicit dependence on the total dimension $d = m \times n$ (which is far larger than the $\sqrt{m+n}$ dependence in our rate).

# B. Basic Properties in Matrix Analysis

We first introduce some basic properties in matrix analysis. For matrices $\mathbf{A}, \mathbf{B} \in \mathbb{R}^{m \times n}$, it holds that

$$\langle \mathbf{A}, \mathbf{B} \rangle = \sum_{i=1}^{m} \sum_{j=1}^{n} \mathbf{A}_{i,j} \mathbf{B}_{i,j} = \text{tr}(\mathbf{A}^T \mathbf{B}),$$

$$\|\mathbf{A}\|_F^2 = \sum_{i=1}^{m} \sum_{j=1}^{n} \mathbf{A}_{i,j}^2 = \text{tr}(\mathbf{A}^T \mathbf{A}),$$

$$\text{tr}(\mathbf{A}^T \mathbf{B}) = \text{tr}(\mathbf{B}\mathbf{A}^T).$$

---

[2]This paper appeared online three months after the submission and first appearance of our paper on arXiv.

**Lemma B.1.** *([Bhatia, 1997](), Lemma V.1.5, Proposition V.1.6) For any matrices $\mathbf{A} \in \mathbb{R}^{m \times s}$ and $\mathbf{X}, \mathbf{Y} \in \mathbb{R}^{m \times m}$ with $\mathbf{X} \preceq \mathbf{Y}$, it holds that*

$$\mathbf{A}^T \mathbf{X} \mathbf{A} \preceq \mathbf{A}^T \mathbf{Y} \mathbf{A}, \quad \text{tr}\,(\mathbf{X}) \leq \text{tr}\,(\mathbf{Y}). \tag{20}$$

*For symmetric positive definite matrices $\mathbf{X}, \mathbf{Y} \in \mathbb{R}^{m \times m}$ with $\mathbf{X} \preceq \mathbf{Y}$, it holds that*

$$\mathbf{X}^{-1} \succeq \mathbf{Y}^{-1}. \tag{21}$$

**Fact B.2.** *For symmetric positive semidefinite matrix $\mathbf{X} \in \mathbb{R}^{m \times m}$, the singular values coincide with the eigenvalues and thus $\text{tr}(\mathbf{X}) = \sum_{i=1}^{m} \sigma_i(\mathbf{X})$.*

**Definition B.3.** For symmetric positive semidefinite matrix $\mathbf{X} \in \mathbb{R}^{m \times m}$, let $\mathbf{U}\mathbf{\Lambda}\mathbf{U}^T$ be its eigenvalue decomposition with $\mathbf{\Lambda} = \text{diag}(\lambda_1, \lambda_2, \cdots, \lambda_m)$, then $\mathbf{X}^p$ is defined to be $\mathbf{U} \, \text{diag}(\lambda_1^p, \lambda_2^p, \cdots, \lambda_m^p)\mathbf{U}^T$.

**Lemma B.4.** *([Ando & Zhan, 1999]()) For symmetric positive semidefinite matrices $\mathbf{X}, \mathbf{Y} \in \mathbb{R}^{m \times m}$ and $0 < p \leq 1$, it holds that*

$$\text{tr}\,((\mathbf{X} + \mathbf{Y})^p) \leq \text{tr}\,(\mathbf{X}^p + \mathbf{Y}^p).$$

*Specially, when $p = 1/2$, we have $\text{tr}\left((\mathbf{X} + \mathbf{Y})^{1/2}\right) \leq \text{tr}\left(\mathbf{X}^{1/2} + \mathbf{Y}^{1/2}\right)$.*

**Lemma B.5.** *For any matrix $\mathbf{X} \in \mathbb{R}^{m \times n}$, it holds that $\|\mathbf{X}\|_* = \text{tr}\left((\mathbf{X}\mathbf{X}^T)^{1/2}\right)$.*

*Proof.* Let $\mathbf{U}\mathbf{\Sigma}\mathbf{V}^T$ be the compact SVD of $\mathbf{X}$ with $r = \text{rank}(\mathbf{X})$, $\mathbf{U} \in \mathbb{R}^{m \times r}$, $\mathbf{\Sigma} \in \mathbb{R}^{r \times r}$, and $\mathbf{V} \in \mathbb{R}^{n \times r}$. Then we have $(\mathbf{X}\mathbf{X}^T)^{1/2} = (\mathbf{U}\mathbf{\Sigma}^2\mathbf{U}^T)^{1/2} = \mathbf{U}(\mathbf{\Sigma}^2)^{1/2}\mathbf{U}^T = \mathbf{U}\mathbf{\Sigma}\mathbf{U}^T$ and $\text{tr}\left((\mathbf{X}\mathbf{X}^T)^{1/2}\right) = \text{tr}\left(\mathbf{U}\mathbf{\Sigma}\mathbf{U}^T\right) = \text{tr}\,(\mathbf{\Sigma}) = \|\mathbf{X}\|_*$. $\qquad\square$

Since $x^p$ with $p \in [0, 1]$ is operator monotonic and operator concave on $[0, \infty)$, we have the following two properties ([Bhatia](), [1997](), Theorem V.1.9, Theorem V.2.5):

**Lemma B.6.** *For symmetric positive semidefinite matrices $\mathbf{X}, \mathbf{Y} \in \mathbb{R}^{m \times m}$ with $\mathbf{X} \preceq \mathbf{Y}$, it holds that $\mathbf{X}^p \preceq \mathbf{Y}^p$ with $p \in [0, 1]$.*

**Lemma B.7.** *For symmetric positive semidefinite matrix $\mathbf{X} \in \mathbb{R}^{m \times m}$, it holds that $\mathbb{E}\,[\mathbf{X}^p] \preceq (\mathbb{E}[\mathbf{X}])^p$ with $p \in [0, 1]$.*

From Assumptions 2 and 3, we have

$$
\begin{aligned}
\mathbb{E}_k\left[\|\mathbf{G}_k - \nabla f(\mathbf{X}_k)\|_F^2 \,\middle|\, \mathcal{F}_{k-1}\right] &= \mathbb{E}_k\left[\text{tr}\left((\mathbf{G}_k - \nabla f(\mathbf{X}_k))(\mathbf{G}_k - \nabla f(\mathbf{X}_k))^T\right) \middle| \mathcal{F}_{k-1}\right] \\
&= \mathbb{E}_k\left[\text{tr}\left((\mathbf{G}_k - \nabla f(\mathbf{X}_k))^T (\mathbf{G}_k - \nabla f(\mathbf{X}_k))\right) \middle| \mathcal{F}_{k-1}\right] \\
&\leq \frac{\text{tr}\,(\Sigma_L) + \text{tr}\,(\Sigma_R)}{2} \equiv \sigma^2,
\end{aligned}
\tag{22}
$$

and

$$
\begin{aligned}
\Sigma_L &\succeq \mathbb{E}_k\left[(\mathbf{G}_k - \nabla f(\mathbf{X}_k))(\mathbf{G}_k - \nabla f(\mathbf{X}_k))^T \middle| \mathcal{F}_{k-1}\right] \\
&= \mathbb{E}_k\left[\mathbf{G}_k\mathbf{G}_k^T + \nabla f(\mathbf{X}_k)\nabla f(\mathbf{X}_k)^T - \mathbf{G}_k\nabla f(\mathbf{X}_k)^T - \nabla f(\mathbf{X}_k)\mathbf{G}_k^T \middle| \mathcal{F}_{k-1}\right] \\
&= \mathbb{E}_k\left[\mathbf{G}_k\mathbf{G}_k^T \middle| \mathcal{F}_{k-1}\right] - \nabla f(\mathbf{X}_k)\nabla f(\mathbf{X}_k)^T.
\end{aligned}
\tag{23}
$$

## C. Proof of Theorem 2.1

*Proof.* As the gradient is $L$-Lipschitz, we have

$$
\begin{aligned}
&\mathbb{E}_k\left[f(\mathbf{X}_{k+1})\middle|\mathcal{F}_{k-1}\right] - f(\mathbf{X}_k) \\
\leq & \mathbb{E}_k\left[\langle \nabla f(\mathbf{X}_k), \mathbf{X}_{k+1} - \mathbf{X}_k \rangle + \frac{L}{2}\|\mathbf{X}_{k+1} - \mathbf{X}_k\|_F^2 \middle| \mathcal{F}_{k-1}\right] \\
= & \mathbb{E}_k\left[-\eta\left\langle \nabla f(\mathbf{X}_k), \lambda\mathbf{X}_k + \mathbf{L}_{k,\varepsilon}^{-\frac{1}{2p}}\mathbf{M}_k\mathbf{R}_{k,\varepsilon}^{-\frac{1}{2q}} \right\rangle + \frac{L\eta^2}{2}\left\|\lambda\mathbf{X}_k + \mathbf{L}_{k,\varepsilon}^{-\frac{1}{2p}}\mathbf{M}_k\mathbf{R}_{k,\varepsilon}^{-\frac{1}{2q}}\right\|_F^2 \middle| \mathcal{F}_{k-1}\right] \\
= & \mathbb{E}_k\left[-\eta\left\langle \mathbf{L}_{k,\varepsilon}^{-\frac{1}{4p}}\nabla f(\mathbf{X}_k)\mathbf{R}_{k,\varepsilon}^{-\frac{1}{4q}}, \lambda\mathbf{L}_{k,\varepsilon}^{\frac{1}{4p}}\mathbf{X}_k\mathbf{R}_{k,\varepsilon}^{\frac{1}{4q}} + \mathbf{L}_{k,\varepsilon}^{-\frac{1}{4p}}\mathbf{M}_k\mathbf{R}_{k,\varepsilon}^{-\frac{1}{4q}} \right\rangle + \frac{L\eta^2}{2}\left\|\lambda\mathbf{X}_k + \mathbf{L}_{k,\varepsilon}^{-\frac{1}{2p}}\mathbf{M}_k\mathbf{R}_{k,\varepsilon}^{-\frac{1}{2q}}\right\|_F^2 \middle| \mathcal{F}_{k-1}\right]
\end{aligned}
\tag{24}
$$

$$
=\mathbb{E}_k\left[-\frac{\eta}{2}\left\|\mathbf{L}_{k,\varepsilon}^{-\frac{1}{4p}}\nabla f(\mathbf{X}_k)\mathbf{R}_{k,\varepsilon}^{-\frac{1}{4q}}\right\|_F^2-\frac{\eta}{2}\left\|\lambda\mathbf{L}_{k,\varepsilon}^{\frac{1}{4p}}\mathbf{X}_k\mathbf{R}_{k,\varepsilon}^{\frac{1}{4q}}+\mathbf{L}_{k,\varepsilon}^{-\frac{1}{4p}}\mathbf{M}_k\mathbf{R}_{k,\varepsilon}^{-\frac{1}{4q}}\right\|_F^2\right.
$$

$$
\left.+\frac{\eta}{2}\left\|\mathbf{L}_{k,\varepsilon}^{-\frac{1}{4p}}\left(\nabla f(\mathbf{X}_k)-\mathbf{M}_k\right)\mathbf{R}_{k,\varepsilon}^{-\frac{1}{4q}}-\lambda\mathbf{L}_{k,\varepsilon}^{\frac{1}{4p}}\mathbf{X}_k\mathbf{R}_{k,\varepsilon}^{\frac{1}{4q}}\right\|_F^2+\frac{L\eta^2}{2}\left\|\lambda\mathbf{X}_k+\mathbf{L}_{k,\varepsilon}^{-\frac{1}{2p}}\mathbf{M}_k\mathbf{R}_{k,\varepsilon}^{-\frac{1}{2q}}\right\|_F^2\Big|\mathcal{F}_{k-1}\right]
$$

$$
\leq\mathbb{E}_k\left[-\frac{\eta}{2}\left\|\mathbf{L}_{k,\varepsilon}^{-\frac{1}{4p}}\nabla f(\mathbf{X}_k)\mathbf{R}_{k,\varepsilon}^{-\frac{1}{4q}}\right\|_F^2-\frac{\eta}{2}\left\|\lambda\mathbf{L}_{k,\varepsilon}^{\frac{1}{4p}}\mathbf{X}_k\mathbf{R}_{k,\varepsilon}^{\frac{1}{4q}}+\mathbf{L}_{k,\varepsilon}^{-\frac{1}{4p}}\mathbf{M}_k\mathbf{R}_{k,\varepsilon}^{-\frac{1}{4q}}\right\|_F^2\right.
$$

$$
\left.+\eta\underbrace{\left\|\mathbf{L}_{k,\varepsilon}^{-\frac{1}{4p}}\left(\nabla f(\mathbf{X}_k)-\mathbf{M}_k\right)\mathbf{R}_{k,\varepsilon}^{-\frac{1}{4q}}\right\|_F^2}_{\text{term (a)}}+\eta\lambda^2\underbrace{\left\|\mathbf{L}_{k,\varepsilon}^{\frac{1}{4p}}\mathbf{X}_k\mathbf{R}_{k,\varepsilon}^{\frac{1}{4q}}\right\|_F^2}_{\text{term (b)}}+\frac{L\eta^2}{2}\underbrace{\left\|\lambda\mathbf{X}_k+\mathbf{L}_{k,\varepsilon}^{-\frac{1}{2p}}\mathbf{M}_k\mathbf{R}_{k,\varepsilon}^{-\frac{1}{2q}}\right\|_F^2}_{\text{term (c)}}\Big|\mathcal{F}_{k-1}\right].
$$

From condition (4) and property (21), we have

$$
\mathbf{L}_{k,\varepsilon}^{-1}\preceq\frac{1}{\hat{\varepsilon}}\mathbf{I}_m,\quad\mathbf{R}_{k,\varepsilon}^{-1}\preceq\frac{1}{\hat{\varepsilon}}\mathbf{I}_n. \tag{25}
$$

For term (a), we have

$$
\begin{aligned}
\left\|\mathbf{L}_{k,\varepsilon}^{-\frac{1}{4p}}\left(\nabla f(\mathbf{X}_k)-\mathbf{M}_k\right)\mathbf{R}_{k,\varepsilon}^{-\frac{1}{4q}}\right\|_F^2 &=\operatorname{tr}\left(\mathbf{R}_{k,\varepsilon}^{-\frac{1}{4q}}\left(\nabla f(\mathbf{X}_k)-\mathbf{M}_k\right)^T\mathbf{L}_{k,\varepsilon}^{-\frac{1}{2p}}\left(\nabla f(\mathbf{X}_k)-\mathbf{M}_k\right)\mathbf{R}_{k,\varepsilon}^{-\frac{1}{4q}}\right)\\
&\overset{a}{\leq}\frac{1}{\hat{\varepsilon}^{\frac{1}{2p}}}\operatorname{tr}\left(\mathbf{R}_{k,\varepsilon}^{-\frac{1}{4q}}\left(\nabla f(\mathbf{X}_k)-\mathbf{M}_k\right)^T\left(\nabla f(\mathbf{X}_k)-\mathbf{M}_k\right)\mathbf{R}_{k,\varepsilon}^{-\frac{1}{4q}}\right)\\
&=\frac{1}{\hat{\varepsilon}^{\frac{1}{2p}}}\operatorname{tr}\left(\left(\nabla f(\mathbf{X}_k)-\mathbf{M}_k\right)\mathbf{R}_{k,\varepsilon}^{-\frac{1}{2q}}\left(\nabla f(\mathbf{X}_k)-\mathbf{M}_k\right)^T\right)\\
&\overset{b}{\leq}\frac{1}{\hat{\varepsilon}^{\frac{1}{2p}+\frac{1}{2q}}}\operatorname{tr}\left(\left(\nabla f(\mathbf{X}_k)-\mathbf{M}_k\right)\left(\nabla f(\mathbf{X}_k)-\mathbf{M}_k\right)^T\right)\\
&\overset{c}{=}\frac{1}{\sqrt{\hat{\varepsilon}}}\left\|\nabla f(\mathbf{X}_k)-\mathbf{M}_k\right\|_F^2,
\end{aligned}
$$

where we use (25), Lemma B.6, and (20) in $\overset{a}{\leq}$ and $\overset{b}{\leq}$, and $\frac{1}{p}+\frac{1}{q}=1$ in $\overset{c}{=}$. For term (b), from the analysis in Section 3.3, we have

$$
\left\|\mathbf{L}_{k,\varepsilon}^{\frac{1}{4p}}\mathbf{X}_k\mathbf{R}_{k,\varepsilon}^{\frac{1}{4q}}\right\|_F^2\leq\frac{9\nu}{\lambda^2 K^{1/2}}\left(\operatorname{tr}\left(\mathbf{L}_{k,\varepsilon}^{1/2}\right)+\operatorname{tr}\left(\mathbf{R}_{k,\varepsilon}^{1/2}\right)\right).
$$

For term (c), similar to the induction for term (a), we have

$$
\begin{aligned}
\left\|\lambda\mathbf{X}_k+\mathbf{L}_{k,\varepsilon}^{-\frac{1}{2p}}\mathbf{M}_k\mathbf{R}_{k,\varepsilon}^{-\frac{1}{2q}}\right\|_F^2 &=\left\|\mathbf{L}_{k,\varepsilon}^{-\frac{1}{4p}}\underbrace{\left(\lambda\mathbf{L}_{k,\varepsilon}^{\frac{1}{4p}}\mathbf{X}_k\mathbf{R}_{k,\varepsilon}^{\frac{1}{4q}}+\mathbf{L}_{k,\varepsilon}^{-\frac{1}{4p}}\mathbf{M}_k\mathbf{R}_{k,\varepsilon}^{-\frac{1}{4q}}\right)}_{\mathbf{A}}\mathbf{R}_{k,\varepsilon}^{-\frac{1}{4q}}\right\|_F^2\\
&=\operatorname{tr}\left(\mathbf{R}_{k,\varepsilon}^{-\frac{1}{4q}}\mathbf{A}^T\mathbf{L}_{k,\varepsilon}^{-\frac{1}{2p}}\mathbf{A}\mathbf{R}_{k,\varepsilon}^{-\frac{1}{4q}}\right)\leq\frac{1}{\hat{\varepsilon}^{\frac{1}{2p}}}\operatorname{tr}\left(\mathbf{R}_{k,\varepsilon}^{-\frac{1}{4q}}\mathbf{A}^T\mathbf{A}\mathbf{R}_{k,\varepsilon}^{-\frac{1}{4q}}\right)\\
&=\frac{1}{\hat{\varepsilon}^{\frac{1}{2p}}}\operatorname{tr}\left(\mathbf{A}\mathbf{R}_{k,\varepsilon}^{-\frac{1}{2q}}\mathbf{A}^T\right)\leq\frac{1}{\hat{\varepsilon}^{\frac{1}{2p}+\frac{1}{2q}}}\operatorname{tr}\left(\mathbf{A}\mathbf{A}^T\right)=\frac{1}{\sqrt{\hat{\varepsilon}}}\|\mathbf{A}\|_F^2\\
&=\frac{1}{\sqrt{\hat{\varepsilon}}}\left\|\lambda\mathbf{L}_{k,\varepsilon}^{\frac{1}{4p}}\mathbf{X}_k\mathbf{R}_{k,\varepsilon}^{\frac{1}{4q}}+\mathbf{L}_{k,\varepsilon}^{-\frac{1}{4p}}\mathbf{M}_k\mathbf{R}_{k,\varepsilon}^{-\frac{1}{4q}}\right\|_F^2.
\end{aligned} \tag{26}
$$

Plugging into (24) and letting $\eta \leq \frac{\sqrt{\hat{\varepsilon}}}{2L}$, we have

$$
\begin{aligned}
&\mathbb{E}_k \left[ f(\mathbf{X}_{k+1}) \big| \mathcal{F}_{k-1} \right] - f(\mathbf{X}_k) \\
&\leq \mathbb{E}_k \left[ -\frac{\eta}{2} \left\| \mathbf{L}_{k,\varepsilon}^{-\frac{1}{4p}} \nabla f(\mathbf{X}_k) \mathbf{R}_{k,\varepsilon}^{-\frac{1}{4q}} \right\|_F^2 - \frac{\eta}{2} \left\| \lambda \mathbf{L}_{k,\varepsilon}^{\frac{1}{4p}} \mathbf{X}_k \mathbf{R}_{k,\varepsilon}^{\frac{1}{4q}} + \mathbf{L}_{k,\varepsilon}^{-\frac{1}{4p}} \mathbf{M}_k \mathbf{R}_{k,\varepsilon}^{-\frac{1}{4q}} \right\|_F^2 + \frac{\eta}{\sqrt{\hat{\varepsilon}}} \left\| \nabla f(\mathbf{X}_k) - \mathbf{M}_k \right\|_F^2 \right. \\
&\quad \left. + \frac{9\eta\nu}{K^{1/2}} \left( \operatorname{tr}\left( \mathbf{L}_{k,\varepsilon}^{1/2} \right) + \operatorname{tr}\left( \mathbf{R}_{k,\varepsilon}^{1/2} \right) \right) + \frac{L\eta^2}{2\sqrt{\hat{\varepsilon}}} \left\| \lambda \mathbf{L}_{k,\varepsilon}^{\frac{1}{4p}} \mathbf{X}_k \mathbf{R}_{k,\varepsilon}^{\frac{1}{4q}} + \mathbf{L}_{k,\varepsilon}^{-\frac{1}{4p}} \mathbf{M}_k \mathbf{R}_{k,\varepsilon}^{-\frac{1}{4q}} \right\|_F^2 \Big| \mathcal{F}_{k-1} \right] \\
&\leq \mathbb{E}_k \left[ -\frac{\eta}{2} \left\| \mathbf{L}_{k,\varepsilon}^{-\frac{1}{4p}} \nabla f(\mathbf{X}_k) \mathbf{R}_{k,\varepsilon}^{-\frac{1}{4q}} \right\|_F^2 - \frac{\eta}{4} \left\| \lambda \mathbf{L}_{k,\varepsilon}^{\frac{1}{4p}} \mathbf{X}_k \mathbf{R}_{k,\varepsilon}^{\frac{1}{4q}} + \mathbf{L}_{k,\varepsilon}^{-\frac{1}{4p}} \mathbf{M}_k \mathbf{R}_{k,\varepsilon}^{-\frac{1}{4q}} \right\|_F^2 \right. \\
&\quad \left. + \frac{\eta}{\sqrt{\hat{\varepsilon}}} \left\| \nabla f(\mathbf{X}_k) - \mathbf{M}_k \right\|_F^2 + \frac{9\eta\nu}{K^{1/2}} \left( \operatorname{tr}\left( \mathbf{L}_{k,\varepsilon}^{1/2} \right) + \operatorname{tr}\left( \mathbf{R}_{k,\varepsilon}^{1/2} \right) \right) \Big| \mathcal{F}_{k-1} \right].
\end{aligned}
\tag{27}
$$

Multiplying both sides of (31) by $\frac{\eta}{\sqrt{\hat{\varepsilon}}(1-\theta)}$, adding it to (27), and arranging the terms, we have

$$
\begin{aligned}
&\mathbb{E}_k \left[ f(\mathbf{X}_{k+1}) - f^* + \frac{\eta}{4} \left\| \lambda \mathbf{L}_{k,\varepsilon}^{\frac{1}{4p}} \mathbf{X}_k \mathbf{R}_{k,\varepsilon}^{\frac{1}{4q}} + \mathbf{L}_{k,\varepsilon}^{-\frac{1}{4p}} \mathbf{M}_k \mathbf{R}_{k,\varepsilon}^{-\frac{1}{4q}} \right\|_F^2 + \frac{\eta\theta}{\sqrt{\hat{\varepsilon}}(1-\theta)} \left\| \nabla f(\mathbf{X}_k) - \mathbf{M}_k \right\|_F^2 \Big| \mathcal{F}_{k-1} \right] \\
&\leq f(\mathbf{X}_k) - f^* + \mathbb{E}_k \left[ -\frac{\eta}{2} \left\| \mathbf{L}_{k,\varepsilon}^{-\frac{1}{4p}} \nabla f(\mathbf{X}_k) \mathbf{R}_{k,\varepsilon}^{-\frac{1}{4q}} \right\|_F^2 + \frac{9\eta\nu}{K^{1/2}} \left( \operatorname{tr}\left( \mathbf{L}_{k,\varepsilon}^{1/2} \right) + \operatorname{tr}\left( \mathbf{R}_{k,\varepsilon}^{1/2} \right) \right) \Big| \mathcal{F}_{k-1} \right] \\
&\quad + \frac{\eta\theta}{\sqrt{\hat{\varepsilon}}(1-\theta)} \left\| \nabla f(\mathbf{X}_{k-1}) - \mathbf{M}_{k-1} \right\|_F^2 + \frac{L^2\eta^3}{\hat{\varepsilon}(1-\theta)^2} \left\| \lambda \mathbf{L}_{k-1,\varepsilon}^{\frac{1}{4p}} \mathbf{X}_{k-1} \mathbf{R}_{k-1,\varepsilon}^{\frac{1}{4q}} + \mathbf{L}_{k-1,\varepsilon}^{-\frac{1}{4p}} \mathbf{M}_{k-1} \mathbf{R}_{k-1,\varepsilon}^{-\frac{1}{4q}} \right\|_F^2 + \frac{\eta(1-\theta)}{\sqrt{\hat{\varepsilon}}} \sigma^2 \\
&\leq f(\mathbf{X}_k) - f^* + \mathbb{E}_k \left[ -\frac{\eta}{2} \left\| \mathbf{L}_{k,\varepsilon}^{-\frac{1}{4p}} \nabla f(\mathbf{X}_k) \mathbf{R}_{k,\varepsilon}^{-\frac{1}{4q}} \right\|_F^2 + \frac{9\eta\nu}{K^{1/2}} \left( \operatorname{tr}\left( \mathbf{L}_{k,\varepsilon}^{1/2} \right) + \operatorname{tr}\left( \mathbf{R}_{k,\varepsilon}^{1/2} \right) \right) \Big| \mathcal{F}_{k-1} \right] \\
&\quad + \frac{\eta\theta}{\sqrt{\hat{\varepsilon}}(1-\theta)} \left\| \nabla f(\mathbf{X}_{k-1}) - \mathbf{M}_{k-1} \right\|_F^2 + \frac{\eta}{4} \left\| \lambda \mathbf{L}_{k-1,\varepsilon}^{\frac{1}{4p}} \mathbf{X}_{k-1} \mathbf{R}_{k-1,\varepsilon}^{\frac{1}{4q}} + \mathbf{L}_{k-1,\varepsilon}^{-\frac{1}{4p}} \mathbf{M}_{k-1} \mathbf{R}_{k-1,\varepsilon}^{-\frac{1}{4q}} \right\|_F^2 + \frac{\eta(1-\theta)}{\sqrt{\hat{\varepsilon}}} \sigma^2,
\end{aligned}
\tag{28}
$$

where we let $\eta^2 \leq \frac{\hat{\varepsilon}(1-\theta)^2}{4L^2}$ in the last inequality. Taking expectation with respect to $\mathcal{F}_{k-1}$ and summing (27) with $k=1$ and (28) over $k = 2, \cdots, K$, we have

$$
\begin{aligned}
&\mathbb{E}_{\mathcal{F}_K} \left[ f(\mathbf{X}_{K+1}) - f^* + \frac{\eta}{4} \left\| \lambda \mathbf{L}_{K,\varepsilon}^{\frac{1}{4p}} \mathbf{X}_K \mathbf{R}_{K,\varepsilon}^{\frac{1}{4q}} + \mathbf{L}_{K,\varepsilon}^{-\frac{1}{4p}} \mathbf{M}_K \mathbf{R}_{K,\varepsilon}^{-\frac{1}{4q}} \right\|_F^2 + \frac{\eta\theta}{\sqrt{\hat{\varepsilon}}(1-\theta)} \left\| \nabla f(\mathbf{X}_K) - \mathbf{M}_K \right\|_F^2 \right] \\
&\leq f(\mathbf{X}_1) - f^* + \sum_{k=1}^K \mathbb{E}_{\mathcal{F}_k} \left[ -\frac{\eta}{2} \left\| \mathbf{L}_{k,\varepsilon}^{-\frac{1}{4p}} \nabla f(\mathbf{X}_k) \mathbf{R}_{k,\varepsilon}^{-\frac{1}{4q}} \right\|_F^2 + \frac{9\eta\nu}{K^{1/2}} \left( \operatorname{tr}\left( \mathbf{L}_{k,\varepsilon}^{1/2} \right) + \operatorname{tr}\left( \mathbf{R}_{k,\varepsilon}^{1/2} \right) \right) \right] \\
&\quad + \left( \frac{\eta\theta}{\sqrt{\hat{\varepsilon}}(1-\theta)} + \frac{\eta}{\sqrt{\hat{\varepsilon}}} \right) \mathbb{E}_{\mathcal{F}_1} \left[ \left\| \nabla f(\mathbf{X}_1) - \mathbf{M}_1 \right\|_F^2 \right] + \frac{(K-1)\eta(1-\theta)}{\sqrt{\hat{\varepsilon}}} \sigma^2 \\
&= f(\mathbf{X}_1) - f^* + \sum_{k=1}^K \mathbb{E}_{\mathcal{F}_k} \left[ -\frac{\eta}{2} \left\| \mathbf{L}_{k,\varepsilon}^{-\frac{1}{4p}} \nabla f(\mathbf{X}_k) \mathbf{R}_{k,\varepsilon}^{-\frac{1}{4q}} \right\|_F^2 + \frac{9\eta\nu}{K^{1/2}} \left( \operatorname{tr}\left( \mathbf{L}_{k,\varepsilon}^{1/2} \right) + \operatorname{tr}\left( \mathbf{R}_{k,\varepsilon}^{1/2} \right) \right) \right] \\
&\quad + \frac{\eta}{\sqrt{\hat{\varepsilon}}(1-\theta)} \mathbb{E}_{\mathcal{F}_1} \left[ \left\| \nabla f(\mathbf{X}_1) - \mathbf{M}_1 \right\|_F^2 \right] + \frac{(K-1)\eta(1-\theta)}{\sqrt{\hat{\varepsilon}}} \sigma^2.
\end{aligned}
\tag{29}
$$

As the gradient is $L$-Lipschitz, we have

$$
f^* \leq f\left( \mathbf{X} - \frac{1}{L}\nabla f(\mathbf{X}) \right) \leq f(\mathbf{X}) - \frac{1}{L} \left\langle \nabla f(\mathbf{X}), \nabla f(\mathbf{X}) \right\rangle + \frac{L}{2} \left\| \frac{1}{L}\nabla f(\mathbf{X}) \right\|_F^2 = f(\mathbf{X}) - \frac{1}{2L} \left\| \nabla f(\mathbf{X}) \right\|_F^2.
$$

Using the recursion of $\mathbf{M}_1$ and $\mathbf{M}_0 = \mathbf{0}$, we have

$$
\begin{aligned}
\mathbb{E}_{\mathcal{F}_1}\left[\|\nabla f(\mathbf{X}_1) - \mathbf{M}_1\|_F^2\right] &= \mathbb{E}_{\mathcal{F}_1}\left[\|\theta \nabla f(\mathbf{X}_1) + (1-\theta)(\nabla f(\mathbf{X}_1) - \mathbf{G}_1)\|_F^2\right] \\
&= \theta^2 \|\nabla f(\mathbf{X}_1)\|_F^2 + (1-\theta)^2 \mathbb{E}_{\mathcal{F}_1}\left[\|\nabla f(\mathbf{X}_1) - \mathbf{G}_1\|_F^2\right] \\
&\leq 2L\left(f(\mathbf{X}_1) - f^*\right) + (1-\theta)^2 \sigma^2.
\end{aligned}
$$

Plugging into (29), we have

$$
\begin{aligned}
&\mathbb{E}_{\mathcal{F}_K}\left[f(\mathbf{X}_{K+1}) - f^* + \frac{\eta}{4}\left\|\lambda \mathbf{L}_{K,\varepsilon}^{\frac{1}{4p}} \mathbf{X}_K \mathbf{R}_{K,\varepsilon}^{\frac{1}{4q}} + \mathbf{L}_{K,\varepsilon}^{-\frac{1}{4p}} \mathbf{M}_K \mathbf{R}_{K,\varepsilon}^{-\frac{1}{4q}}\right\|_F^2 + \frac{\eta\theta}{\sqrt{\hat{\varepsilon}}(1-\theta)}\|\nabla f(\mathbf{X}_K) - \mathbf{M}_K\|_F^2\right] \\
&\leq f(\mathbf{X}_1) - f^* + \sum_{k=1}^K \mathbb{E}_{\mathcal{F}_k}\left[-\frac{\eta}{2}\left\|\mathbf{L}_{k,\varepsilon}^{-\frac{1}{4p}}\nabla f(\mathbf{X}_k)\mathbf{R}_{k,\varepsilon}^{-\frac{1}{4q}}\right\|_F^2 + \frac{9\eta\nu}{K^{1/2}}\left(\operatorname{tr}\left(\mathbf{L}_{k,\varepsilon}^{1/2}\right) + \operatorname{tr}\left(\mathbf{R}_{k,\varepsilon}^{1/2}\right)\right)\right] \\
&\quad + \frac{2L\eta}{\sqrt{\hat{\varepsilon}}(1-\theta)}\left(f(\mathbf{X}_1) - f^*\right) + \frac{K\eta(1-\theta)}{\sqrt{\hat{\varepsilon}}}\sigma^2
\end{aligned}
$$

and

$$
\begin{aligned}
&\sum_{k=1}^K \mathbb{E}_{\mathcal{F}_k}\left[\left\|\mathbf{L}_{k,\varepsilon}^{-\frac{1}{4p}}\nabla f(\mathbf{X}_k)\mathbf{R}_{k,\varepsilon}^{-\frac{1}{4q}}\right\|_F^2\right] \\
&\leq \frac{18\nu}{K^{1/2}}\sum_{k=1}^K \mathbb{E}_{\mathcal{F}_k}\left[\operatorname{tr}\left(\mathbf{L}_{k,\varepsilon}^{1/2}\right) + \operatorname{tr}\left(\mathbf{R}_{k,\varepsilon}^{1/2}\right)\right] + \frac{2\left(f(\mathbf{X}_1) - f^*\right)}{\eta} + \frac{4L}{\sqrt{\hat{\varepsilon}}(1-\theta)}\left(f(\mathbf{X}_1) - f^*\right) + \frac{2K(1-\theta)}{\sqrt{\hat{\varepsilon}}}\sigma^2 \\
&\leq \frac{18\nu}{K^{1/2}}\sum_{k=1}^K \mathbb{E}_{\mathcal{F}_k}\left[\operatorname{tr}\left(\mathbf{L}_{k,\varepsilon}^{1/2}\right) + \operatorname{tr}\left(\mathbf{R}_{k,\varepsilon}^{1/2}\right)\right] + \underbrace{\frac{2\left(f(\mathbf{X}_1) - f^*\right)}{\eta} + \frac{4L}{\sqrt{\hat{\varepsilon}}(1-\theta)}\left(f(\mathbf{X}_1) - f^*\right) + \frac{2K(1-\theta)}{\sqrt{\hat{\varepsilon}}}\hat{\sigma}^2}_{C'},
\end{aligned}
$$

where we denote $\hat{\sigma}^2 = \max\left\{\sigma^2, \frac{L(f(\mathbf{X}_1) - f^*)}{K\gamma^2}\right\}$ with any $\gamma \in (0, 1]$. From Lemma 3.4, we have

$$
\sum_{k=1}^K \mathbb{E}\left[\operatorname{tr}\left(\mathbf{L}_{k,\varepsilon}^{1/2}\right) + \operatorname{tr}\left(\mathbf{R}_{k,\varepsilon}^{1/2}\right)\right] \leq K\left(\underbrace{\operatorname{tr}\left(\Sigma_L^{1/2}\right) + \operatorname{tr}\left(\Sigma_R^{1/2}\right) + (m+n)\sqrt{\varepsilon}}_{C}\right) + \frac{4}{\sqrt{1-\beta}}\sum_{t=1}^K \mathbb{E}\left[\|\nabla f(\mathbf{X}_t)\|_*\right].
$$

From Lemma 3.3, we have

$$
\begin{aligned}
&\sum_{k=1}^K \mathbb{E}\left[\|\nabla f(\mathbf{X}_k)\|_*\right] \\
&\leq \sqrt{\left(\sum_{k=1}^K \mathbb{E}\left[\operatorname{tr}\left(\mathbf{L}_{k,\varepsilon}^{1/2}\right) + \operatorname{tr}\left(\mathbf{R}_{k,\varepsilon}^{1/2}\right)\right]\right)\left(\sum_{k=1}^K \mathbb{E}\left[\left\|\mathbf{L}_{k,\varepsilon}^{-\frac{1}{4p}}\nabla f(\mathbf{X}_k)\mathbf{R}_{k,\varepsilon}^{-\frac{1}{4q}}\right\|_F^2\right]\right)} \\
&\leq \sqrt{\left(\sum_{k=1}^K \mathbb{E}\left[\operatorname{tr}\left(\mathbf{L}_{k,\varepsilon}^{1/2}\right) + \operatorname{tr}\left(\mathbf{R}_{k,\varepsilon}^{1/2}\right)\right]\right)\left(\frac{18\nu}{K^{1/2}}\sum_{k=1}^K \mathbb{E}\left[\operatorname{tr}\left(\mathbf{L}_{k,\varepsilon}^{1/2}\right) + \operatorname{tr}\left(\mathbf{R}_{k,\varepsilon}^{1/2}\right)\right] + C'\right)} \\
&\leq \sqrt{\left(\frac{4}{\sqrt{1-\beta}}\sum_{t=1}^K \mathbb{E}\left[\|\nabla f(\mathbf{X}_t)\|_*\right] + KC\right)\left(\frac{18\nu}{K^{1/2}}\frac{4}{\sqrt{1-\beta}}\sum_{t=1}^K \mathbb{E}\left[\|\nabla f(\mathbf{X}_t)\|_*\right] + 18\nu C K^{1/2} + C'\right)}.
\end{aligned}
$$

So we have

$$
\left(\sum_{k=1}^{K} \mathbb{E}\left[\|\nabla f(\mathbf{X}_k)\|_*\right]\right)^2
$$

$$
\leq \frac{288\nu}{K^{1/2}(1-\beta)}\left(\sum_{k=1}^{K}\mathbb{E}\left[\|\nabla f(\mathbf{X}_k)\|_*\right]\right)^2 + \frac{4}{\sqrt{1-\beta}}\left(36\nu C K^{1/2} + C'\right)\sum_{k=1}^{K}\mathbb{E}\left[\|\nabla f(\mathbf{X}_k)\|_*\right] + 18\nu C^2 K^{3/2} + KC'C. \tag{30}
$$

Next, we consider the constants. From Definition B.3 and Fact B.2, we have

$$
\operatorname{tr}\left(\Sigma_L^{1/2}\right) + \operatorname{tr}\left(\Sigma_R^{1/2}\right) = \sum_{i=1}^{m}\sqrt{\sigma_i(\Sigma_L)} + \sum_{i=1}^{n}\sqrt{\sigma_i(\Sigma_R)} \leq \sqrt{(m+n)\left(\sum_{i=1}^{m}\sigma_i(\Sigma_L) + \sum_{i=1}^{n}\sigma_i(\Sigma_R)\right)}
$$

$$
= \sqrt{(m+n)\left(\operatorname{tr}\left(\Sigma_L\right) + \operatorname{tr}\left(\Sigma_R\right)\right)} = \sigma\sqrt{2(m+n)} \leq \hat{\sigma}\sqrt{2(m+n)}.
$$

Letting $\varepsilon = \frac{\tau\hat{\sigma}^2}{m+n}$ with any $\tau \leq 1$, we have $(m+n)\sqrt{\varepsilon} \leq \hat{\sigma}\sqrt{m+n}$ and

$$
C \leq 2.5\hat{\sigma}\sqrt{m+n}.
$$

Recall that we require the parameters satisfying the following relations in the above proof

$$
\eta \leq \frac{\sqrt{\hat{\varepsilon}}}{2L}, \quad \eta^2 \leq \frac{\hat{\varepsilon}(1-\theta)^2}{4L^2}
$$

and

$$
\eta\lambda \leq \frac{\sqrt{\nu}}{2K^{5/4}}, \quad \|\mathbf{X}_1\|_{op} \leq \frac{\sqrt{\nu}}{K^{1/4}\lambda}, \quad \frac{\sqrt{\nu}}{K^{1/4}} \leq 1, \quad \theta \leq \beta \leq \sqrt{\theta} < 1
$$

in Lemma 3.8. Letting

$$
1-\theta = \sqrt{\frac{L\left(f(\mathbf{X}_1) - f^*\right)}{K\hat{\sigma}^2}}, \quad \eta = \sqrt{\frac{\hat{\varepsilon}\left(f(\mathbf{X}_1) - f^*\right)}{4LK\hat{\sigma}^2}}, \quad \nu = \frac{1}{1152}\sqrt{\frac{L(f(\mathbf{X}_1) - f^*)}{\hat{\sigma}^2}},
$$

$$
\lambda \leq \frac{1}{\sqrt{1152\hat{\varepsilon}}K^{3/4}}\sqrt[4]{\frac{L^3\hat{\sigma}^2}{f(\mathbf{X}_1) - f^*}}, \quad \|\mathbf{X}_1\|_{op} \leq \sqrt{\frac{\hat{\varepsilon}K\left(f(\mathbf{X}_1) - f^*\right)}{L\hat{\sigma}^2}},
$$

the above requirements are satisfied by the definition of $\hat{\sigma}^2$. We also have

$$
\frac{1}{1-\beta} \leq \frac{1}{1-\sqrt{\theta}} \leq \frac{2}{1-\theta} = 2\sqrt{\frac{K\hat{\sigma}^2}{L\left(f(\mathbf{X}_1) - f^*\right)}}, \quad \frac{288\nu}{K^{1/2}(1-\beta)} \leq \frac{1}{2}, \quad C' \leq 10\sqrt{\frac{K\hat{\sigma}^2 L\left(f(\mathbf{X}_1) - f^*\right)}{\hat{\varepsilon}}},
$$

$$
\frac{C'}{\sqrt{1-\beta}} \leq \frac{14.2\hat{\sigma}}{\sqrt{\hat{\varepsilon}}}\sqrt[4]{K^3\hat{\sigma}^2 L\left(f(\mathbf{X}_1) - f^*\right)}, \quad \frac{\nu C K^{1/2}}{\sqrt{1-\beta}} \leq \frac{\sqrt{m+n}}{288}\sqrt[4]{K^3\hat{\sigma}^2 L\left(f(\mathbf{X}_1) - f^*\right)},
$$

$$
\frac{4}{\sqrt{1-\beta}}\left(36\nu C K^{1/2} + C'\right) \leq \left(\sqrt{m+n} + \frac{57\hat{\sigma}}{\sqrt{\hat{\varepsilon}}}\right)\sqrt[4]{K^3\hat{\sigma}^2 L\left(f(\mathbf{X}_1) - f^*\right)},
$$

$$
\nu C^2 K^{3/2} \leq \frac{m+n}{144}\sqrt{K^3\hat{\sigma}^2 L\left(f(\mathbf{X}_1) - f^*\right)}, \quad KC'C \leq 25\hat{\sigma}\sqrt{\frac{m+n}{\hat{\varepsilon}}}\sqrt{K^3\hat{\sigma}^2 L\left(f(\mathbf{X}_1) - f^*\right)},
$$

$$
18\nu C^2 K^{3/2} + KC'C \leq \left(m+n+25\hat{\sigma}\sqrt{\frac{m+n}{\hat{\varepsilon}}}\right)\sqrt{K^3\hat{\sigma}^2 L(f(\mathbf{X}_1) - f^*)} \leq \left(\frac{27}{2}(m+n) + \frac{25}{2}\frac{\hat{\sigma}^2}{\hat{\varepsilon}}\right)\sqrt{K^3\hat{\sigma}^2 L(f(\mathbf{X}_1) - f^*)}.
$$

So we have

$$
\frac{1}{2}\left(\sum_{k=1}^{K}\mathbb{E}\left[\|\nabla f(\mathbf{X}_k)\|_*\right]\right)^2 \leq \left(\sqrt{m+n} + \frac{57\hat{\sigma}}{\sqrt{\hat{\varepsilon}}}\right)\sqrt[4]{K^3\hat{\sigma}^2 L\left(f(\mathbf{X}_1) - f^*\right)}\sum_{k=1}^{K}\mathbb{E}\left[\|\nabla f(\mathbf{X}_k)\|_*\right]
$$

$$
+ \left(\frac{27}{2}(m+n) + \frac{25}{2}\frac{\hat{\sigma}^2}{\hat{\varepsilon}}\right)\sqrt{K^3\hat{\sigma}^2 L\left(f(\mathbf{X}_1) - f^*\right)}.
$$

Solving inequality $x^2 - ax - b \leq 0$, we have $x \leq \frac{a + \sqrt{a^2 + 4b}}{2} \leq a + \sqrt{b}$ and

$$\sum_{k=1}^{K} \mathbb{E}\left[\|\nabla f(\mathbf{X}_k)\|_*\right] \leq \left(2\sqrt{m+n} + \frac{114\hat{\sigma}}{\sqrt{\hat{\varepsilon}}} + \sqrt{27(m+n) + 25\frac{\hat{\sigma}^2}{\hat{\varepsilon}}}\right) \sqrt[4]{K^3\hat{\sigma}^2 L\left(f(\mathbf{X}_1) - f^*\right)}$$

$$\leq \left(8\sqrt{m+n} + \frac{119\hat{\sigma}}{\sqrt{\hat{\varepsilon}}}\right) \sqrt[4]{K^3\hat{\sigma}^2 L\left(f(\mathbf{X}_1) - f^*\right)}$$

$$= \left(8\sqrt{m+n} + \frac{119\hat{\sigma}}{\sqrt{\hat{\varepsilon}}}\right) \max\left\{\sqrt[4]{K^3\sigma^2 L\left(f(\mathbf{X}_1) - f^*\right)}, \sqrt{\frac{KL\left(f(\mathbf{X}_1) - f^*\right)}{\gamma}}\right\}.$$

Dividing both sides by $K$, we have the conclusion. At last, Lemma 3.8 guarantees

$$\lambda\|\mathbf{X}_k\|_{op} \leq \frac{3\sqrt{\nu}}{K^{1/4}} = \frac{3}{\sqrt{1152}}\sqrt[4]{\frac{L\left(f(\mathbf{X}_1) - f^*\right)}{K\hat{\sigma}^2}} < 1$$

by the setting of $\hat{\sigma}^2$. $\hfill\square$

## D. Supporting Lemmas

The following lemma extends (Li et al., 2025a, Lemma 3) but replaces the $\ell_\infty$ norm of vectors by the spectral norm of matrices.

**Lemma D.1.** *Let* $\eta\lambda \leq \frac{\sqrt{\nu}}{2K^{5/4}}$, $\|\mathbf{X}_1\|_{op} \leq \frac{\sqrt{\nu}}{K^{1/4}\lambda}$, $\frac{\sqrt{\nu}}{K^{1/4}} \leq 1$, $\theta \leq \beta \leq \sqrt{\theta} < 1$, *and* $\frac{1}{p} + \frac{1}{q} = 1$. *Then for Algorithm 1, we have*

$$\lambda\|\mathbf{X}_k\|_{op} \leq \frac{3\sqrt{\nu}}{K^{1/4}}, \quad \forall k = 1, 2, \cdots, K.$$

*Proof.* From the update of $\mathbf{X}_{k+1}$, we have

$$\|\mathbf{X}_{k+1}\|_{op} - \frac{2}{\lambda} = \left\|(1 - \lambda\eta)\mathbf{X}_k - \eta\mathbf{L}_{k,\varepsilon}^{-\frac{1}{2p}}\mathbf{M}_k\mathbf{R}_{k,\varepsilon}^{-\frac{1}{2q}}\right\|_{op} - \frac{2}{\lambda}$$

$$\leq (1 - \lambda\eta)\|\mathbf{X}_k\|_{op} + \eta\left\|\mathbf{L}_{k,\varepsilon}^{-\frac{1}{2p}}\mathbf{M}_k\mathbf{R}_{k,\varepsilon}^{-\frac{1}{2q}}\right\|_{op} - \frac{2}{\lambda}$$

$$\overset{a}{\leq} (1 - \lambda\eta)\|\mathbf{X}_k\|_{op} + 2\eta - \frac{2}{\lambda}$$

$$= (1 - \lambda\eta)\left(\|\mathbf{X}_k\|_{op} - \frac{2}{\lambda}\right)$$

$$\leq (1 - \lambda\eta)^k\left(\|\mathbf{X}_1\|_{op} - \frac{2}{\lambda}\right)$$

$$= -\frac{1}{\lambda}(1 - \lambda\eta)^k\left(2 - \frac{\sqrt{\nu}}{K^{1/4}}\right),$$

where we use Lemma 3.6 in $\overset{a}{\leq}$. Since $\ln x \leq x - 1$ and $e^x \geq x + 1$ for any $x > 0$ and $\eta\lambda \leq \frac{\sqrt{\nu}}{2K^{5/4}} \leq \frac{1}{2}$, we have for any $k \leq K$ that

$$k\ln(1 - \eta\lambda) = -k\ln\frac{1}{1 - \eta\lambda} \geq -K\left(\frac{1}{1 - \eta\lambda} - 1\right) = -\frac{K\eta\lambda}{1 - \eta\lambda} \geq -\frac{\sqrt{\nu}}{K^{1/4}},$$

$$(1 - \eta\lambda)^k \geq e^{-\frac{\sqrt{\nu}}{K^{1/4}}} \geq 1 - \frac{\sqrt{\nu}}{K^{1/4}},$$

and

$$\|\mathbf{X}_{k+1}\|_{op} - \frac{2}{\lambda} \leq -\frac{1}{\lambda}\left(1 - \frac{\sqrt{\nu}}{K^{1/4}}\right)\left(2 - \frac{\sqrt{\nu}}{K^{1/4}}\right) \leq -\frac{2}{\lambda} + \frac{3}{\lambda}\frac{\sqrt{\nu}}{K^{1/4}}.$$

$\hfill\square$

The following lemma is closely similar to (Li et al., 2025a, Lemma 4) and we list the proof here only for the sake of completeness.

**Lemma D.2.** *Suppose that Assumptions 1-3 and condition (4) hold and let $\frac{1}{p} + \frac{1}{q} = 1$. Then for Algorithm 1, we have*

$$
\begin{aligned}
&\mathbb{E}_k \left[ \|\mathbf{M}_k - \nabla f(\mathbf{X}_k)\|_F^2 \,\big| \mathcal{F}_{k-1} \right] \\
&\leq \theta \|\mathbf{M}_{k-1} - \nabla f(\mathbf{X}_{k-1})\|_F^2 + \frac{L^2 \eta^2}{(1-\theta)\sqrt{\hat\varepsilon}} \left\| \lambda \mathbf{L}_{k-1,\varepsilon}^{\frac{1}{4p}} \mathbf{X}_{k-1} \mathbf{R}_{k-1,\varepsilon}^{\frac{1}{4q}} + \mathbf{L}_{k-1,\varepsilon}^{-\frac{1}{4p}} \mathbf{M}_{k-1} \mathbf{R}_{k-1,\varepsilon}^{-\frac{1}{4q}} \right\|_F^2 + (1-\theta)^2 \sigma^2.
\end{aligned}
\tag{31}
$$

*Proof.* Denoting $\Gamma_k = \mathbf{G}_k - \nabla f(\mathbf{X}_k)$, we have $\mathbb{E}_k \left[ \Gamma_k | \mathcal{F}_{k-1} \right] = 0$ and $\mathbb{E}_k \left[ \|\Gamma_k\|_F^2 | \mathcal{F}_{k-1} \right] \leq \sigma^2$ from (22). From the update of $\mathbf{M}_k$, we have

$$
\begin{aligned}
\mathbf{M}_k - \nabla f(\mathbf{X}_k) =& \theta \mathbf{M}_{k-1} + (1-\theta)\mathbf{G}_k - \nabla f(\mathbf{X}_k) \\
=& \theta \left( \mathbf{M}_{k-1} - \nabla f(\mathbf{X}_{k-1}) \right) + (1-\theta)\left( \nabla f(\mathbf{X}_k) + \Gamma_k \right) - \nabla f(\mathbf{X}_k) + \theta \nabla f(\mathbf{X}_{k-1}) \\
=& \theta \left( \mathbf{M}_{k-1} - \nabla f(\mathbf{X}_{k-1}) \right) + (1-\theta)\Gamma_k - \theta \left( \nabla f(\mathbf{X}_k) - \nabla f(\mathbf{X}_{k-1}) \right)
\end{aligned}
$$

and

$$
\begin{aligned}
&\mathbb{E}_k \left[ \|\mathbf{M}_k - \nabla f(\mathbf{X}_k)\|_F^2 \,\big| \mathcal{F}_{k-1} \right] \\
&= \|\theta \left( \mathbf{M}_{k-1} - \nabla f(\mathbf{X}_{k-1}) \right) - \theta \left( \nabla f(\mathbf{X}_k) - \nabla f(\mathbf{X}_{k-1}) \right)\|_F^2 + (1-\theta)^2 \mathbb{E}_k \left[ \|\Gamma_k\|_F^2 \,\big| \mathcal{F}_{k-1} \right] \\
&\leq \theta^2 \left( 1 + \frac{1-\theta}{\theta} \right) \|\mathbf{M}_{k-1} - \nabla f(\mathbf{X}_{k-1})\|_F^2 + \theta^2 \left( 1 + \frac{\theta}{1-\theta} \right) \|\nabla f(\mathbf{X}_k) - \nabla f(\mathbf{X}_{k-1})\|_F^2 + (1-\theta)^2 \mathbb{E}_k \left[ \|\Gamma_k\|_F^2 \,\big| \mathcal{F}_{k-1} \right] \\
&\leq \theta \|\mathbf{M}_{k-1} - \nabla f(\mathbf{X}_{k-1})\|_F^2 + \frac{1}{1-\theta} \|\nabla f(\mathbf{X}_k) - \nabla f(\mathbf{X}_{k-1})\|_F^2 + (1-\theta)^2 \sigma^2 \\
&\leq \theta \|\mathbf{M}_{k-1} - \nabla f(\mathbf{X}_{k-1})\|_F^2 + \frac{L^2}{1-\theta} \|\mathbf{X}_k - \mathbf{X}_{k-1}\|_F^2 + (1-\theta)^2 \sigma^2 \\
&= \theta \|\mathbf{M}_{k-1} - \nabla f(\mathbf{X}_{k-1})\|_F^2 + \frac{L^2 \eta^2}{1-\theta} \left\| \lambda \mathbf{X}_{k-1} + \mathbf{L}_{k-1,\varepsilon}^{-\frac{1}{2p}} \mathbf{M}_{k-1} \mathbf{R}_{k-1,\varepsilon}^{-\frac{1}{2q}} \right\|_F^2 + (1-\theta)^2 \sigma^2 \\
&\leq \theta \|\mathbf{M}_{k-1} - \nabla f(\mathbf{X}_{k-1})\|_F^2 + \frac{L^2 \eta^2}{(1-\theta)\sqrt{\hat\varepsilon}} \left\| \lambda \mathbf{L}_{k-1,\varepsilon}^{\frac{1}{4p}} \mathbf{X}_{k-1} \mathbf{R}_{k-1,\varepsilon}^{\frac{1}{4q}} + \mathbf{L}_{k-1,\varepsilon}^{-\frac{1}{4p}} \mathbf{M}_{k-1} \mathbf{R}_{k-1,\varepsilon}^{-\frac{1}{4q}} \right\|_F^2 + (1-\theta)^2 \sigma^2,
\end{aligned}
$$

where we use (26) in the last inequality. $\qquad\square$

**Lemma D.3.** *When each element of $\mathbf{G} \in \mathbb{R}^{m \times n}$ is generated from Gaussian distribution with mean $\mu$ and variance $\xi^2$ independently, we have*

$$
\mathbb{E} \left[ \mathbf{G}\mathbf{G}^T \right] \succeq n\xi^2 \mathbf{I}_m = \frac{\xi^2}{m(\xi^2 + \mu^2)} \mathbb{E} \left[ \|\mathbf{G}\|_F^2 \right] \mathbf{I}_m,
$$

$$
\mathbb{E} \left[ \mathbf{G}^T\mathbf{G} \right] \succeq m\xi^2 \mathbf{I}_n = \frac{\xi^2}{n(\xi^2 + \mu^2)} \mathbb{E} \left[ \|\mathbf{G}\|_F^2 \right] \mathbf{I}_n.
$$

*Proof.* When $\mathbf{G}_{i,j} \sim \mathcal{N}(\mu, \xi^2)$, we have

$$
\mathbb{E} \left[ \left( \mathbf{G}\mathbf{G}^T \right)_{p,q} \right] = \mathbb{E} \left[ \sum_{j=1}^n \mathbf{G}_{p,j} \mathbf{G}_{q,j} \right] = \sum_{j=1}^n \mathbb{E}[\mathbf{G}_{p,j}\mathbf{G}_{q,j}] = \sum_{j=1}^n \mathbb{E}[\mathbf{G}_{p,j}] \, \mathbb{E} \left[ \mathbf{G}_{q,j} \right] = n\mu^2 \quad \text{if} \quad p \neq q
$$

and

$$
\mathbb{E} \left[ \left( \mathbf{G}\mathbf{G}^T \right)_{p,q} \right] = \sum_{j=1}^n \mathbb{E} \left[ \mathbf{G}_{p,j}^2 \right] = \sum_{j=1}^n \left( \mathbb{E} \left[ (\mathbf{G}_{p,j} - \mu)^2 \right] + \mu^2 \right) = n(\xi^2 + \mu^2) \quad \text{if} \quad p = q.
$$

So

$$\mathbb{E}\left[\mathbf{G}\mathbf{G}^T\right] = n\mu^2\mathbf{1}_m\mathbf{1}_m^T + n\xi^2\mathbf{I}_m \succeq n\xi^2\mathbf{I}_m,$$

where $\mathbf{1}_m \in \mathbb{R}^m$ is the vector with all ones. We also have

$$\mathbb{E}\left[\|\mathbf{G}\|_F^2\right] = \sum_{i=1}^m\sum_{j=1}^n \mathbb{E}\left[\mathbf{G}_{i,j}^2\right] = \sum_{i=1}^m\sum_{j=1}^n\left(\mathbb{E}\left[(\mathbf{G}_{i,j}-\mu)^2\right] + \mu^2\right) = mn(\xi^2+\mu^2).$$

So we have

$$\mathbb{E}\left[\mathbf{G}\mathbf{G}^T\right] \succeq n\xi^2\mathbf{I}_m = \frac{\xi^2}{m(\xi^2+\mu^2)}\mathbb{E}\left[\|\mathbf{G}\|_F^2\right]\mathbf{I}_m.$$

Similarly, we also have

$$\mathbb{E}\left[\mathbf{G}^T\mathbf{G}\right] \succeq m\xi^2\mathbf{I}_n = \frac{\xi^2}{n(\xi^2+\mu^2)}\mathbb{E}\left[\|\mathbf{G}\|_F^2\right]\mathbf{I}_n.$$

$\square$

# E. Experiments

In this section, we conduct experiments on real-world deep learning tasks to examine whether the theoretical claims developed in the main paper are reflected in practical training dynamics. Specifically, we examine the two kinds of relationships that make our convergence rate practically meaningful: the nuclear-to-Frobenius norm ratio of the full gradient and the effective spectral floor $\hat{\varepsilon}$ of the Shampoo preconditioners relative to the noise-dependent scale $\hat{\sigma}^2/(m+n)$ and the numerical regularizer $\varepsilon$. We pretrain the GPT-2 (Radford et al., 2019) model from scratch on the OpenWebText dataset (Gokaslan et al., 2019) to validate these claims. [3]

A central quantity in our empirical evaluation is the full gradient $\nabla f(\mathbf{X})$, which is required both for computing the gradient norm ratio and for estimating the stochastic gradient noise $\sigma^2$. Following the protocol of Li et al. (2025a, Section E), we alternately switch between *training phases* and *logging phases*, and use the logging phases to periodically approximate $\nabla f(\mathbf{X})$ via accumulated large batch gradients. This interleaved design allows us to measure the full gradient related quantities needed in the analysis without interfering with the normal training dynamics.

Following common practice in non-diagonal preconditioning methods (Jordan et al., 2024; Anil et al., 2020; Shi et al., 2023), we apply the AdamW-style Shampoo optimizer only to non-embedding two-dimensional parameters, while using AdamW for the remaining parameters. Concretely, for a Transformer block (omitting the multi-head structure) the attention and feed-forward modules can be written as

$$\text{Attn}(\mathbf{X}) = \text{softmax}\left(\frac{\mathbf{X}\mathbf{W}_Q(\mathbf{X}\mathbf{W}_K)^\top}{\sqrt{d}}\right)\mathbf{X}\mathbf{W}_V\mathbf{W}_O,$$

and

$$\text{MLP}(\mathbf{X}) = \phi(\mathbf{X}\mathbf{W}_1)\mathbf{W}_2,$$

where $\phi$ denotes the activation function. The Shampoo-optimized parameters therefore consist of four representative matrix classes: (i) the attention input projection matrices, corresponding to the merged QKV projection in GPT-2, i.e. $\text{concat}(\mathbf{W}_Q\mathbf{W}_K\mathbf{W}_V)$; (ii) the attention output projection matrix $\mathbf{W}_O$; (iii) the first layer in MLP module $\mathbf{W}_1$; and (iv) the second layer in MLP module $\mathbf{W}_2$. For clarity and brevity, we report the averaged results within each class.

For the training recipe, we use the standard Megatron-LM GPT-2 Small configuration (Shoeybi et al., 2019) and follow the related works for shampoo settings (Anil et al., 2020; Shi et al., 2023) with minimal modifications. The model contains 12 Transformer layers, hidden size 768, 12 attention heads, and sequence length 1024. We set the learning rate to $10^{-2}$ and weight decay to 0.05. For the Shampoo-preconditioned parameters, we set $\theta = 0.9$, $\beta = 0.999$, $\varepsilon = 10^{-12}$, and $p = q = 2$; for the remaining parameters handled by the AdamW, we use $(\beta_1, \beta_2) = (0.9, 0.95)$. The learning rate is linearly warmed

---

[3]Our code is available at `https://github.com/adonis-dym/Convergence-Rate-AdamW-Style-Shampoo`.

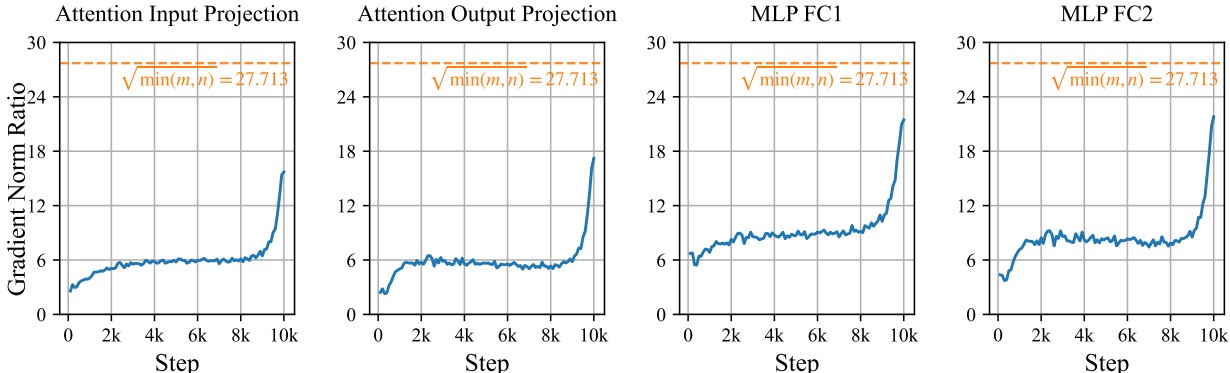

*Figure 2.* The gradient norm ratio $\|\nabla f(\mathbf{X})\|_* / \|\nabla f(\mathbf{X})\|_F$ during GPT-2 pretraining on OpenWebText for each class of Shampoo-handled parameters. The dashed horizontal line indicates the value of $\sqrt{\min(m,n)}$ for each matrix shape.

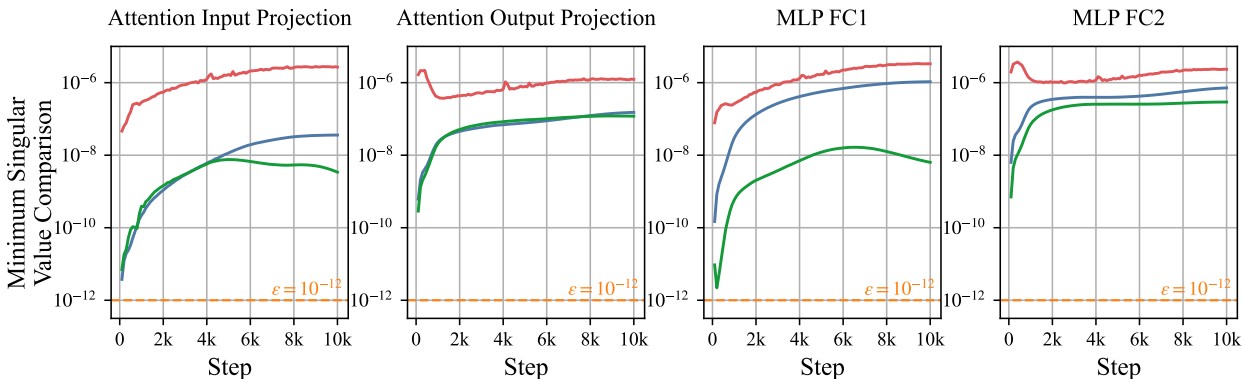

*Figure 3.* Comparison of the minimum eigenvalues with respect to the stochastic gradient noise during GPT-2 pretraining on OpenWebText dataset. The red curve indicates the noise-dependent scale $\hat{\sigma}^2/(m+n)$, while the blue and green curves correspond to the minimum eigenvalues of the regularized left and right preconditioner matrices $\mathbf{L}_{k,\varepsilon}$ and $\mathbf{R}_{k,\varepsilon}$, respectively. All quantities are plotted on a logarithmic scale.

up over the first 1000 training steps and then decayed according to a cosine schedule over the remaining 9000 steps. We set the global batch size to 640 and apply gradient clipping with threshold 1.0, and complete the training with 8 NVIDIA H200 GPUs.

Figure 2 reports the value of the gradient norm ratio $\|\nabla f(\mathbf{X})\|_* / \|\nabla f(\mathbf{X})\|_F$ for the four classes of Shampoo-handled matrices. Across all parameter categories, we observe that this ratio stays at the same magnitude as the theoretical upper bound $\sqrt{\min(m,n)}$ throughout training, indicated by the dashed reference lines. This behavior suggests that the relationship $\|\nabla f(\mathbf{X})\|_* = \Theta(\sqrt{\min\{m,n\}}) \, \|\nabla f(\mathbf{X})\|_F$ indeed holds in practice.

Next, we turn to the empirical estimation of the stochastic gradient noise, and compare the noise-dependent scale $\hat{\sigma}^2/(m+n)$ with $\varepsilon$ and $\hat{\varepsilon}$. To make the noise-dependent scale tractable, we use the approximation $\hat{\sigma}^2 \approx \sigma^2 \approx \|\mathbf{G} - \nabla f(\mathbf{X})\|_F^2$, where $\mathbf{G}$ is the mini-batch stochastic gradient and $\nabla f(\mathbf{X})$ is approximated by the accumulated large batch gradient from the logging phase. We then compute the corresponding scale $\hat{\sigma}^2/(m+n)$ for each Shampoo-handled matrix. To estimate the effective spectral floor $\hat{\varepsilon}$, we record the minimum eigenvalues of the regularized left and right preconditioner matrices $\mathbf{L}_{k,\varepsilon}$ and $\mathbf{R}_{k,\varepsilon}$, respectively. These quantities are readily available during training, since they are produced as a byproduct of the eigendecomposition used to compute the matrix inverse roots in Algorithm 1.

We show the results in Figure 3. Across all four classes of Shampoo-handled parameters, the minimum eigenvalues of both $\mathbf{L}_{k,\varepsilon}$ and $\mathbf{R}_{k,\varepsilon}$ are consistently several orders of magnitude larger than the numerical regularizer $\varepsilon = 10^{-12}$. Moreover, these estimates of $\hat{\varepsilon}$ remain comparable to the noise-dependent scale $\hat{\sigma}^2/(m+n)$ throughout training. This indicates that the

theoretical bound (5) derived in Theorem 2.1 aligns much closer to the favorable regime in (7) than the worse-case rate (6).

### E.1. Comparison with the Other Optimizers

The AdamW-style Shampoo, originally implemented in (Shi et al., 2023), is a classical optimizer that won the AlgoPerf neural network training competition (Kasimbeg et al., 2025). It has been adopted as a baseline for comparison in the literature. For instance, Frans et al. (2025a) compared different optimizers on GPT-2 using the OpenWebText dataset. Table 2 of (Frans et al., 2025a) shows that Shampoo (in the AdamW-style) outperforms AdamW and achieves performance comparable to Muon (Jordan et al., 2024). SOAP (Vyas et al., 2025) and Splus (Frans et al., 2025b), which are both extensions of Shampoo, achieve even better performance in (Frans et al., 2025a). Therefore, we omit comparisons with other optimizers in this theoretical paper.

