# OpenReview forum: "Convergence Rate Analysis of the AdamW-Style Shampoo: Unifying One-Sided and Two-Sided Preconditioning"
_ICML.cc/2026/Conference — ICML 2026 regular_

### Official Review · Reviewer_qPF8 · 2026-03-06

**Soundness:** 3
**Presentation:** 3
**Significance:** 3
**Originality:** 3
**Overall Recommendation:** 4
**Confidence:** 3

**Summary:**

This paper analyzes the convergence of an AdamW-style Shampoo optimizer for nonconvex matrix optimization, combining (i) momentum, (ii) decoupled weight decay, and (iii) one- or two-sided non-diagonal preconditioning. The main guarantee is a nonconvex convergence bound measured via the expected average \emph{nuclear norm} of the gradient, with a rate that (up to a $\sqrt{m+n}$ factor) matches the optimal SGD rate in favorable regimes. The paper also proves an implicit bias/boundedness result in spectral norm, drawing an analogy to AdamW's $\ell_1/\ell_\infty$ behavior but lifted to matrix singular values.

The analysis treats both two-sided Shampoo (finite $p,q$ such as $p=q=2$) and one-sided variants (one of $p,q=\infty$ with $1/p+1/q=1$) in a unified framework, using Schatten-$p$ H\"older inequalities as a key technical tool. The paper additionally addresses a practical mismatch between theory and implementation regarding tiny $\varepsilon$ values by motivating an effective $\hat{\varepsilon}$ (and a condition ensuring preconditioners are not dominated by $\varepsilon$). Empirical results on GPT-2 pretraining (OpenWebText) are used to validate (a) the observed ratio $\|\nabla f(X)\|_\ast / \|\nabla f(X)\|_F$ and (b) the eigen-scale of preconditioners relative to $\varepsilon$.

**Compliance With Llm Reviewing Policy:**

Affirmed.

**Final Justification:**

I thank the authors for the clarification provided during the rebuttal phase. After careful consideration of the response and the subsequent discussion, my overall evaluation of the paper remains the same.

**Key Questions For Authors:**

1. Under what conditions should the nuclear-norm stationarity measure be viewed as a reliable proxy for the more standard Frobenius-norm notion?
2. How conservative is the upper bound required on the weight decay parameter?
 3. Can the analysis be adapted (even with weaker constants) to more standard training regimes with fixed $(\beta,\theta)$ and scheduled learning rates?
 4. Is there a practical way to estimate or track the relevant effective preconditioner scale online during training?

**Limitations:**

Yes.

**Strengths And Weaknesses:**

The paper’s main strength is its technical contribution. Non-convex convergence analysis for practically relevant Shampoo variants is difficult, and the treatment of two-sided preconditioning is particularly valuable. The paper also analyzes a relatively realistic optimizer, including momentum and decoupled weight decay, rather than an overly simplified variant. The proof strategy appears sophisticated and well tailored to the matrix-valued setting.

The main weakness is that the result is somewhat indirect. The convergence guarantee is stated in nuclear norm rather than Frobenius norm, and the interpretation as an analogue of the optimal SGD rate depends on additional assumptions about the relation between these quantities. The appendix provides some empirical support, but this does not fully resolve the concern that the main bound may be difficult to interpret operationally.

A second limitation is that the analysis relies on several nontrivial assumptions and parameter restrictions, including lower bounds on the regularized preconditioners and a sufficiently small weight decay parameter. The paper does address these issues, but it remains unclear how restrictive they are in realistic training settings and which are essential versus proof artifacts.

The empirical section is also limited. The experiments support certain theoretical surrogates, but they do not directly demonstrate improved optimization behavior or training performance. For a paper motivated by practical Shampoo methods, some direct optimization evidence would strengthen the case considerably.

---

> ### Author Rebuttal · Authors · 2026-03-28
>
> We sincerely thank the reviewers for their constructive feedback and address the key concerns below.
>
> $\textbf{Weakness 1 and Question 1: Nuclear norm and Frobenius norm}$
>
> Due to limited characters, please see the response to $\textit{Weakness 2 and Question 2 from Reviewer G8A9}$.
>
> $\textbf{Weakness 2 and Question 2: Small weight decay parameter and lower bounds on the regularized preconditioners}$
>
> The theoretical setting for the small weight decay parameter $\lambda$ is employed solely to derive the worst-case bound and is not intended for direct practical use. In practice, it is conservative: using larger $\lambda$ often does not degrade empirical training performance. For example, Shi et al. (2023) adopt $\lambda=10^{-4}$ in their implementation, which exceeds the value required by our theoretical setting. On the other hand, large values of $\lambda$ are typically avoided in practice, as they may drive the parameters toward zero and away from the minimum solution. Therefore, while our theoretical settings are somewhat conservative, they still reflect the trends observed in practice.
>
> Please see the response to $\textit{Weakness 3 from Reviewer G8A9}$ for the lower bounds on the regularized preconditioners. The above reasoning also applies to the parameter $\epsilon$.
>
> $\textbf{Weakness 3: Training performance comparison (also to Weakness 4 and Question 3 from Reviewer G8A9 and Weakness 4 from Reviewer uB4b)}$
>
> The AdamW‑style Shampoo, originally implemented in (Shi et al., 2023), is a classical optimizer that won the AlgoPerf neural network training competition (Kasimbeg et al., 2025). It has been adopted as a baseline for comparison in many literature. For instance, Reference [1] conducts experiments on GPT‑2 using the OpenWebText dataset. Table 2 of [1] presents a comparison among different optimizers. We summarize the validation loss below.
>
> |          | AdamW | Shampoo | Splus | SOAP | Muon |
> |--------------|----|-----|-----|-----|-----|
> | Val Loss | 2.982 | 2.963 | 2.954 | 2.9392 | 2.964 |
>
> We see that Shampoo (in the AdamW-style) outperforms AdamW and achieves performance comparable to Muon. SOAP and Splus, which are both extensions of Shampoo, achieve even better performance.
>
> Additionally, we validate our findings on the Qwen model in a production-scale setting by pretraining Qwen3.5-35B-A3B (an open-source LLM) on 100B tokens and comparing AdamW, Muon, SOAP, one-sided Shampoo, and two-sided Shampoo under matched configurations. Our implementation of Shampoo largely follows (Shi et al., 2023) and is integrated with the ZeRO-1 distributed optimization strategy, enabling accelerated training while preserving mathematical equivalence.
>
> The experimental results show that two-sided Shampoo achieves the lowest training loss and yields the best performance. Due to limited characters, we report the training losses of two-sided Shampoo and the second-best optimizer (Muon) below:
>
> | Step | 1 | 2500 | 5000 | 7500 | 10000 | 12500 |
> |------|---|------|------|------|-------|--------|
> | **Two-sided Shampoo** | 12.613 | 2.296 | 2.096 | 2.019 | 1.902 | 1.846 |
> | **Muon** | 12.613 | 2.308 | 2.043 | 1.970 | 1.905 | 1.880 |
>
> The full loss curves for all optimizers are available at the anonymous link (permitted by ICML) https://anonymous.4open.science/r/convergence-rate-shampoo/qwen_optimizer_lm_loss.pdf.
>
> Overall, these results suggest that two-sided Shampoo is both theoretically grounded and practically effective in LLM training scenarios.
>
> [1]. What Really Matters in Matrix-Whitening Optimizers? arXiv: 2510.25000, 2025.
>
> $\textbf{Question 3}$
>
> Our analysis does not extend to regimes with constant, iteration‑independent values of $(\theta,\beta)$. Taking Adam as the example, prior work [2,3] proves through constructed examples that it fails to converge to stationary points with the common hyperparameter settings of $\theta=0.9$ and $\beta=0.999, 0.997, 0.995, 0.993$. See [3, Figure 2].
>
> Regarding scheduled learning rates, we conjecture that the analysis can be adapted to the case $\eta_k=O(1/\sqrt{k})$. See [2, Theorem 3.1] for example. However, adaptation to the commonly used cosine decay scheduler may not be feasible.
>
> [2]. Adam can converge without any modification on update rules. NeurIPS 2022.
>
> [3]. Provable adaptivity of Adam under non-uniform smoothness. KDD 2024.
>
> $\textbf{Question 4}$
>
> We infer that the reviewer's term 'preconditioner scale' refers to $L^{-1/p}$ and $R^{-1/q}$. (Shi et al., 2023, Section 3.2.1) described two matrix root inverse solvers. One is based on symmetric eigendecomposition: $L^{-1/p}=U\Lambda^{-1/p}U^T$ with $L=U\Lambda U^T$ being the eigendecomposition. The other employs coupled inverse Newton iteration: $X_{t+1}=X_t\left(\frac{(p+1)I-M_t}{p}\right)$, $M_{t+1}=\left(\frac{(p+1)I-M_t}{p}\right)^pM_t$ with $M_0=\frac{1}{c^p}L$, then one expects $X_t\rightarrow L^{-1/p}$. $X_0$ can be initialized to $L_{k-1}^{-1/p}$ from previous iteration to save computation.

---

> > ### Author Rebuttal · Reviewer_qPF8 · 2026-04-01
> >
> > We thank the authors for their response and our recommendation for weak accept remains.

---

### Official Review · Reviewer_uB4b · 2026-03-08

**Soundness:** 3
**Presentation:** 3
**Significance:** 3
**Originality:** 3
**Overall Recommendation:** 4
**Confidence:** 3

**Summary:**

The prior papers have analyzed the convergence rate in the convex case, and the other prior paper analyzed the convergence rate of a one-sided variant of Shampoo in the non-convex case. However, no existing papers have analyzed the two-sided variant of Shampoo in the non-convex setting. This paper provides the convergence analysis of (both the one-sided and two-sided versions of) Shampoo in the non-convex setting. Then, this paper shows that Shampoo may converge faster than SGD since $\| \nabla f (X) \|_\star$ is bounded instead of $\| \nabla f (X) \|_F$.

**Compliance With Llm Reviewing Policy:**

Affirmed.

**Key Questions For Authors:**

See the weakness section.

**Limitations:**

yes

**Strengths And Weaknesses:**

### Strengths
- This paper provides the convergence analysis of (the two-sided version of) Shampoo in the non-convex setting.
- This paper shows that Shampoo can achieve the same convergence rate as SGD in the worst case, while Shampoo may converge faster than SGD since $\| \nabla f (x) \|_\star \leq \| \nabla f (X) \|_F$.

### Weaknesses
- It seems that the convergence rate shown in Theorem 2.1 does not depend on $p$ and $q$. Is it possible to discuss the advantage of two-sided preconditioning over one-sided preconditioning from Theorem 2.1?
- This paper uses Assumption 3, which is a bit stronger than the usual assumption: $\| \nabla f (X_k) - G_k \|_F^2 \leq \sigma^2$. However, the results shown in Theorem 2.1 do not depend on $\Sigma_L$ and $\Sigma_R$. Is it necessary to use Assumption 3?
- Convergence rate shown in Eq. (1) is a bit strange since it becomes zero when $\sigma=0$. I think $\sqrt{\frac{L (f (X_1) - f^\star)}{K}}$ is missing.
- There are no numerical results in this paper. It would be better if this paper evaluates Shampoo and SGD and shows that Shampoo can converge faster than SGD in the synthetic function. It would be helpful to verify the results shown in this paper that Shampoo may converge faster than SGD.

---

> ### Author Rebuttal · Authors · 2026-03-29
>
> We sincerely thank the reviewers for their constructive feedback and address the key concerns below.
>
> $\textbf{Weakness 1}$
>
> 1. We use the relation $\frac{1}{p}+\frac{1}{q}=1$ together with matrix inequalities to relax $p$ and $q$ to the constant 1 in our proofs. Consequently, Theorem 2.1 does not depend on $p$ and $q$.
>
> 2. Theorem 2.1 establishes the same convergence rate for both one‑sided and two‑sided Shampoo. From the perspective of convergence rate, we do not observe a theoretical advantage of the two‑sided variant, as we conjecture that the rate derived in Theorem 2.1 may be the tightest possible and it applies to both settings. Please refer to our responses to $\textit{Weakness 2 and Question 2 from Reviewer G8A9}$ for a discussion on the possible tightness of the result.
>
> 3. Intuitively, the left preconditioning $LG$ captures correlations within each column of $G$, while the right preconditioning $GR$ captures correlations within each row of $G$. Two‑sided preconditioning combines these advantages and captures correlations within both rows and columns of $G$.
>
> $\textbf{Weakness 2}$
>
> 1. The convergence rate derived in inequality (24) (lines 770–777) depends on $\Sigma_L$ and $\Sigma_R$ rather than on $\sigma^2$. However, to simplify the final convergence rate and align the bound with that of SGD, we relax $tr(\Sigma_L^{1/2})+tr(\Sigma_R^{1/2})$ to $\sigma\sqrt{2(m+n)}$ on lines 779–783. Consequently, the final result stated in Theorem 2.1 does not depend on $\Sigma_L$ or $\Sigma_R$​.
>
> 2. Assumption 3 is essential in the proof of Lemma 3.4 (see lines 300–303, left column) for establishing the tight convergence rate. We conjecture that the same convergence rate cannot be derived under the standard bounded variance assumption for the Shampoo-class methods.
>
> $\textbf{Weakness 3}$
>
> Thanks for this suggestion. We describe only the convergence rate of SGD for the stochastic optimization setting in (1) and omit the deterministic case for brevity. The rigorous rate is $max\left(\sqrt[4]{\frac{\sigma^2L\triangle_f}{K}},\sqrt{\frac{L\triangle_f}{K}}\right)$.
>
> $\textbf{Weakness 4}$
>
> Following reference [1], we consider the matrix quadratic regression problem: $min_{X}\frac{1}{2N}\sum_{i=1}^N ||A(i)XB(i)-C(i)||_F^2$, with $X\in R^{20\times10}$, $A(i)\in R^{20\times20}$, $B(i)\in R^{10\times10}$, and $N=1000$. Each element of $A(i), B(i), C(i)$ is drawn from a Gaussian distribution. We compare AdamW‑style Shampoo with SGD. The batch size is set to $10$, the weight decay parameter to $0.01$ for both methods, and $(\theta,\beta,\epsilon)=(0.9,0.99,10^{-8})$ for AdamW-style Shampoo. The optimal learning rates are tuned to $10^{-6}$ for SGD and $10^{-5}$ for AdamW-style Shampoo (larger learning rates degrade performance for both methods). The loss function values and gradient Frobenius norms are listed below, and we observe that AdamW‑style Shampoo converges faster.
>
> - Loss function:
> | Epoch         | 10 | 20 | 30 | 40 | 50 |
> |--------------|----|-----|-----|-----|-----|
> | SGD |   99.7369 |  99.7353 |  99.7353 | 99.7352 |   99.7353 |
> | Shampoo | 99.7426 |  99.7354 |  99.7349 | 99.7348 | 99.7348 |
>
> - Gradient Frobenius norms:
> | Epoch         | 10 | 20 | 30 | 40 | 50 |
> |--------------|----|-----|-----|-----|-----|
> | SGD |   1.0296 |  0.6464 |  0.6682 | 0.6347 |   0.6488 |
> | Shampoo | 1.8202 | 0.6736 |  0.5332 | 0.4624 | 0.4827 |
>
> Please refer to our responses to $\textit{Weakness 3 from Reviewer qPF8}$ for more experimental comparisons on Qwen.
>
> [1]. PolarGrad: A Class of Matrix-Gradient Optimizers from a Unifying Preconditioning Perspective, arXiv: 2505.21799, 2025.

---

> > ### Author Rebuttal · Reviewer_uB4b · 2026-04-01
> >
> > Most of my concerns have been addressed.
> > Since my original score is already positive, I would like to keep my score.

---

### Official Review · Reviewer_LtwF · 2026-03-11

**Soundness:** 3
**Presentation:** 1
**Significance:** 3
**Originality:** 3
**Overall Recommendation:** 4
**Confidence:** 4

**Summary:**

The paper studies the minimization of matrix-valued objective functions, a setting that naturally arises in many practical machine learning applications, such as neural network layers. In this context, a wide range of optimizers has been proposed. Recently, some of the most popular approaches have been steepest descent methods in non-Euclidean geometries equipped with adaptive preconditioners, including AdamW, Shampoo, SOAP, Muon, and various hybrids thereof. While the practical performance of these methods has been extensively studied, their convergence guarantees are less understood, especially in the nonconvex setting.

The paper provides a unified convergence analysis for a broad class of preconditioned descent methods. The framework encompasses left-, right-, and two-sided gradient preconditioning, allows for variable powers of the preconditioner matrices, and incorporates momentum and decoupled weight decay. This unification is technically meaningful and clarifies the relationships between several practically important algorithms.

**Compliance With Llm Reviewing Policy:**

Affirmed.

**Final Justification:**

I would like to thank the authors for answering my questions in rebuttal. I also read other reviews and rebuttals, and would like to retain my recommendation -- _acceptance, conditional on a revision of the presentation and related work discussion. (I am not entirely sure whether ICML has a formal mechanism to ensure such revisions are implemented.)_

**Key Questions For Authors:**

I have two main questions for the authors:
1. Can you explicitly list the algorithms that fall within your framework?

2. For those algorithms, how does your convergence analysis compare to the guarantees in their original papers (e.g., in terms of assumptions, rates, constants, or generality)?

**Limitations:**

Yes.

**Strengths And Weaknesses:**

Personally, I find the analysis very interesting, and I believe it will appeal to a broad segment of the machine learning theory and optimization communities. One of the strengths of the paper is that it confirms that many seemingly different adaptive algorithms indeed share convergence guarantees comparable to SGD. Some of the intermediate results are also insightful. For example, the bound on the update norm (by a constant factor of 2) establishes a connection to Signed Gradient Descent and Normalized Gradient Descent, which is an interesting observation.

That said, I believe the literature review is inadequate. There are many related works that were not mentioned. In particular, there are works that generalize right/left preconditioning or elementwise adaptive updates using generalized “preconditioned” norms and non-Euclidean steepest descent frameworks. Also, listing which algorithms can be interpreted as special cases of the proposed framework would strengthen the paper. Additionaly, popular broader literature should also be discussed to properly situate the contribution.

I would suggest authors to check following papers:

- Carlson, David E., et al. "Preconditioned spectral descent for deep learning." Advances in neural information processing systems 28 (2015).

- Carlson, David, et al. "Stochastic spectral descent for discrete graphical models." IEEE Journal of Selected Topics in Signal Processing 10.2 (2015): 296-311.

- Veprikov, Andrey, et al. "Preconditioned Norms: A Unified Framework for Steepest Descent, Quasi-Newton and Adaptive Methods." arXiv preprint arXiv:2510.10777 (2025).

- Crawshaw, Michael, et al. "An exploration of non-euclidean gradient descent: Muon and its many variants." arXiv preprint arXiv:2510.09827 (2025).

- Shazeer, Noam, and Mitchell Stern. "Adafactor: Adaptive learning rates with sublinear memory cost." International conference on machine learning. PMLR, 2018.

- Reddi, Sashank J., Satyen Kale, and Sanjiv Kumar. "On the convergence of adam and beyond." arXiv preprint arXiv:1904.09237 (2019).

- You, Yang, Igor Gitman, and Boris Ginsburg. "Large batch training of convolutional networks." arXiv preprint arXiv:1708.03888 (2017).

- You, Yang, et al. "Large batch optimization for deep learning: Training bert in 76 minutes." arXiv preprint arXiv:1904.00962 (2019).

In addition, the presentation requires improvement. The paper uses a lot of notation, and some terms are not formally defined. Multiple constants or symbols appear before being properly introduced. Certain passages are difficult to follow due to imprecise wording (e.g., the last paragraph of Section 1.1). There are also some confusing referencing -- for instance, numbers in bracket “(1)” are refering to two distinct equations/inequalities at the same time.

Overall, I believe that the results are interesting and worth publishing. However, the presentation and literature positioning require revision. Therefore, I would recommend acceptance, conditional on a revision of the presentation and related work discussion. (I am not entirely sure whether ICML has a formal mechanism to ensure such revisions are implemented.)

---

> ### Author Rebuttal · Authors · 2026-03-27
>
> We sincerely thank the reviewers for their constructive feedback and address the key concerns below.
>
> $\textbf{Weakness 1: Literature review on preconditioned non-Euclidean steepest descent methods}$
>
> 1. Algorithmic design perspective
>
> Reference [1] investigates the non-Euclidean gradient descent with the update:
>
> $x_{k+1}=argmin_{x} \left< g_k,x-x_k \right> + \frac{1}{2\gamma}||x-x_k||^2\qquad\qquad (1)$
>
> and its preconditioned variant
>
> $x_{k+1}=argmin_{x} \left< g_k,x-x_k \right> + \frac{1}{2\gamma}||x-x_k||_{D_k}^2\qquad\qquad (2)$
>
> It is well known that Shampoo is a special case of (2) when the norm is the Euclidean $\ell_2$​ norm and we set $x_k=vec(X_k)$, $g_k=vec(G_k)$, and $D_k=(R_k\otimes L_k)^{1/4}$. Here, $\otimes$ denotes the Kronecker product and we utilize the identity $vec(L_k^{-1/4}G_kR_k^{-1/4})=(R_k\otimes L_k)^{-1/4}vec(G_k)$.
>
> Reference [2] proposes the stochastic gradient descent in non-Euclidean space, which is a special case of iterate (1).
>
> Reference [4] studies the constrained non-Euclidean steepest descent, which is the trust-region counterpart of iterate (1):
>
> $x_{k+1}=argmin_{||x-x_k||\leq \eta} \left< g_k,x-x_k \right>=x_k+\eta LMO_{||\cdot||}(g_k)\qquad\qquad (3)$
>
> where the linear minimization oracle (LMO) is defined as $LMO_{||\cdot||}(g)=argmin_{||u||=1}\left< g_k,u \right>$. By choosing different norms, (3) recovers normalized gradient descent, SignSGD, and Muon as special cases.
>
> Reference [3] further considers preconditioning and presents the most comprehensive algorithmic framework, encompassing many well-known methods. For the diagonal preconditioning (referred to as adaptive methods in [3]), the following update is investigated:
>
> $X_{k+1}=X_k+\eta D_k^{\circ -1}\odot LMO_{||\cdot||}(D_k^{\circ -1}\odot G_k)\qquad\qquad (4)$
>
> where $\odot$ denotes the Hadamard product and $D^{\circ -1}$ represents the element-wise inversion.  For the non-diagonal preconditioning (termed quasi Newton methods in [3]), the update is:
>
> $X_{k+1}=X_k+\eta L_k^{-1} LMO_{||\cdot||}(L_k^{-1}G_kR_k^{-1})R_k^{-1}\qquad\qquad (5)$
>
> References [5–8] all study specific instances of diagonally preconditioned (or element-wise preconditioned) methods, including Adafactor, AmsGrad, LARS, and LAMB. These methods do not include Shampoo nor fall within the Shampoo class.
>
> 2. Convergence analysis perspective
>
> From a theoretical standpoint, reference [2] establishes convergence guarantees for (1) in both convex and nonconvex settings. However, $\textbf{regarding preconditioned cases, }$ $\textbf{including (2), (4), and (5) with dynamic preconditioners, }$ $\textbf{there is no convergence studies in references [1–8].}$ While many existing methods can be unified under the framework of preconditioned non-Euclidean steepest descent from an algorithmic design perspective, a unified convergence analysis remains challenging due to the complicated and dynamic nature of the preconditioners. To the best of our knowledge, no such analysis exists in the literature.
>
> We thank the reviewer for pointing out these references. In our revised manuscript, we will enhance the literature review about the unified preconditioned non-Euclidean steepest descent framework. However, specifically focusing on the topic of $\textit{convergence analysis for the Shampoo-class methods}$, an area that our literature review has emphasized, existing literature remains scarce, and we therefore consider our literature review to be adequate.
>
> $\textbf{Weakness 2: Presentation}$
>
> We thank the reviewer for this constructive suggestion. Given that $\textbf{ICML permits one additional page in the camera-ready version to incorporate revisions }$ $\textbf{from the rebuttal}$, we will have sufficient space to enhance the overall presentation.
>
> $\textbf{Question 1: Algorithms that fall within our framework}$
>
> Due to the distinction between diagonal and non-diagonal preconditioners, diagonally preconditioned methods, including AdaGrad, RMSProp, Adam, AdamW, Adafactor, AmsGrad, LARS, and LAMB, do not fall within our framework. In contrast, our framework subsumes non-diagonally preconditioned methods, with the original Shampoo and one-sided Shampoo as direct special cases.
>
> $\textbf{Question 2: Convergence comparison to the original papers of those algorithms}$
>
> In the original Shampoo paper, convergence is established only for online convex optimization (lines 87–90, left column). In contrast, the literature on nonconvex convergence analysis remains scarce. To the best of our knowledge, (Xie et al., 2025b) appears to be the only related work. Different to our paper, (Xie et al., 2025b) focuses on one-sided Shampoo in the AdaGrad-style and RMSProp-style, and their analysis does not incorporate momentum or decoupled weight decay. See lines 97–105 (left column) for a detailed review of (Xie et al., 2025b), and lines 203–219 (right column) for a comparison including assumptions, convergence rates, and constants.

---

> > ### Author Rebuttal · Reviewer_LtwF · 2026-04-07
> >
> > I would like to thank the authors for answering my questions in rebuttal. I also read other reviews and rebuttals, and would like to retain my recommendation -- _acceptance, conditional on a revision of the presentation and related work discussion. (I am not entirely sure whether ICML has a formal mechanism to ensure such revisions are implemented.)_

---

### Official Review · Reviewer_G8A9 · 2026-03-13

**Soundness:** 3
**Presentation:** 2
**Significance:** 2
**Originality:** 2
**Overall Recommendation:** 4
**Confidence:** 3

**Summary:**

The paper studies the nonconvex convergence theory of AdamW-style Shampoo that combines exponential moving average momentum, decoupled weight decay, and matrix preconditioning, while also covering both two-sided preconditioning and one-sided preconditioning in a single framework.

They consider a unified framework that recover classical two-sided Shampoo as a special case and also include one-sided variants when one exponent is infinite.

The main theoretical claim is a nonconvex convergence rate of the form $\frac{1}{K}\sum_{k=1}^K \mathbb{E}|\nabla f(X_k)|_*\le O\left(\frac{\sqrt{m+n} C}{K^{1/4}}\right)$.

**Compliance With Llm Reviewing Policy:**

Affirmed.

**Key Questions For Authors:**

1. Why do the authors adopt the left/right bounded noise variance assumptions in Lines 123–127, instead of the more standard condition $$\mathbb{E}\big[|G_t-\nabla f(X_t)|_F^2\big]\le \sigma^2$$?
  Are both of these matrix-sided inequalities truly essential for the analysis, or could the results be established under a simpler Frobenius-norm variance bound?

2. The main convergence guarantee is stated in terms of the nuclear norm. However, the nuclear norm differs from the Frobenius norm only up to a dimension-dependent constant factor. Could the authors better clarify why a nuclear-norm stationarity result is important here, and what additional insight it provides beyond a more standard Frobenius-norm guarantee?

3. Since practical AdamW-style Shampoo is motivated in part by empirical performance, it would be helpful to include direct optimization comparisons against strong baselines such as AdamW, one-sided Shampoo, SOAP, or Muon on a language-model pretraining task. Why are such comparisons absent from the experimental section?

4. The analysis imposes the coupling condition
  $$\theta \le \beta \le \sqrt{\theta}$$.
 Is this constraint mainly a technical artifact of the proof, or does it reflect parameter choices that are meaningful in practice? It would be useful if the authors could clarify whether this condition is theoretically necessary or empirically motivated.

**Strengths And Weaknesses:**

**Strengths**

S1. The problem is important. Shampoo-style optimizers are practically relevant, and the paper targets a theory gap for the AdamW-style implementation.

S2. The unification of one-sided and two-sided preconditioning is meaningful. Prior nonconvex Shampoo-related theory, as described by the paper, focuses mostly on one-sided variants, whereas this work directly handles the more difficult two-sided case together with momentum and decoupled weight decay.

S3. The proof route is nontrivial and reasonably interesting. The use of Schatten-norm Holder inequalities and matrix Cauchy–Schwarz appears well matched to the matrix structure of the optimizer and goes beyond a straightforward lifting of scalar AdamW arguments.

S4. The technical core of the proof is to handle the difficulty of two-sided matrix preconditioning. The paper develops bounds based on **Schatten-(p) Holder inequalities** to control the nuclear norm of the gradient, and a **matrix Cauchy–Schwarz inequality** to control the operator norm of the preconditioned momentum term. These ingredients are then combined with a descent analysis and a recursion for the momentum tracking error.

S5. The theoretical rate is cleanest when $\tau=1$, but this is acknowledged as impractical, and the authors introduce condition (3) to partially bridge that theory-practice mismatch.


**Weaknesses**

W1. The paper has limited algorithmic novelty and only moderate theoretical novelty.

W2. The main guarantee is in nuclear norm rather than Frobenius norm. The analogy to SGD is therefore conditional on a favorable relation between these norms, and the extra (\sqrt{m+n}) factor is not fully eliminated. This weakens the sharpness of the result relative to classical nonconvex first-order rates.

W3. A substantial part of the practical relevance hinges on condition (3), namely uniform lower bounds on the preconditioners by (\hat\varepsilon I). While the paper gives heuristic justification and some empirical support, this condition is additional and materially affects the final rate constants.

W4. The empirical section is too narrow for a top-tier ML conference if the paper wants to claim practical significance. The GPT-2 study mainly validates theoretical surrogates; it does not convincingly demonstrate optimization or generalization advantages over AdamW, one-sided Shampoo, SOAP, or Muon under controlled budgets.

---

> ### Author Rebuttal · Authors · 2026-03-28
>
> We sincerely thank the reviewers for their constructive feedback and address the key concerns below.
>
> $\textbf{Weakness 1}$
>
> Our paper analyzes a classic and highly influential optimizer in the history of deep learning optimization, and we emphasize that filling the theoretical gap for such a classical algorithm is highly meaningful. Specifically, we introduce novel proof techniques and establish tight convergence rates. We believe this theoretical grounding offers meaningful value to the community.
>
> $\textbf{Weakness 2 and Question 2 (also to Weakness 1 and Question 1 from Reviewer qPF8)}$
>
> 1. Insight of nuclear norm rather than Frobenius norm
>
> SGD, SignSGD, and Muon correspond to $\textit{non-Euclidean steepest descent}$ under the $\ell_2$, $\ell_{\infty}$, and spectral norms, respectively, and their convergence rates are measured by the $\ell_2$, $\ell_1$, and nuclear norms [1, 2] (see our response to $\textit{Weakness 1 from Reviewer LtwF}$). If we ignore the accumulation in the preconditioners, Shampoo reduces to Muon (An et al., 2025, Proposition 1 in Section G). This relationship is analogous to that between AdaGrad and SignSGD. Therefore, it is reasonable to employ the nuclear norm—as is done for Muon—to measure the convergence of Shampoo.
>
> Shampoo generalizes AdaGrad from vectors to matrices, and AdaGrad employs $\ell_1$ norm to measure the convergence. Consider the following simplified scenario: if $m=n$ and during the iteration process, the left and right singular matrices $(U_k,V_k)$ of $G_k$ remain unchanged, then $L_k^{-1/4}G_kR_k^{-1/4}=U\left(\sum_{t=1}^k\Sigma_t^2\right)^{-1/4}\Sigma_k\left(\sum_{t=1}^k\Sigma_t^2\right)^{-1/4}V^T$ and Shampoo is equivalent to running AdaGrad on the singular values. This also supports the use of nuclear norm in Shampoo since it serves as a natural matrix extension of the $\ell_1$ norm applied to singular values.
>
> 2. The $O(\sqrt{m+n})$ factor
>
> According to the recent work [3, Theorem 3.1], the lower bound for AdaGrad is $E[min_{1\leq t\leq T}||\nabla f(x_t)||_1]=O\left(\frac{d^2\sigma^2L\left(f(x_1)-f^*\right)\log T}{T}\right)^{1/4}$ under standard assumptions of $L$-smoothness and $\sigma^2$-bounded variance, where $d$ denotes the dimension. Since Shampoo generalizes AdaGrad, and given the unavoidable $\sqrt{d}$​ dependence in AdaGrad’s lower bound, we conjecture that the convergence rate derived in our paper is sharp and that the $O(\sqrt{m+n})$ factor cannot be eliminated.
>
> 3. Why nuclear norm is important
>
> The advantage of nuclear norm is that it yields tighter convergence rate using current techniques. It is technically challenging to eliminate the $O(\sqrt{m+n})$ factor and achieve the exact same convergence rate as SGD when using the Frobenius norm; consequently, the resulting convergence rate would remain persistently $O(\sqrt{m+n})$ times slower than that of SGD. In contrast, employing the nuclear norm allows the convergence rate to become analogous to that of SGD in the ideal scenario.
>
> When $||\nabla f(X)||_*=\Theta(\sqrt{\min(m,n)})||\nabla f(X)||_F$, we think nuclear norm measure is a reliable proxy for the Frobenius norm.
>
> [1]. SIGNSGD: Compressed Optimisation for Non-Convex Problems. ICML 2018.
>
> [2]. On the Convergence Analysis of Muon, arXiv 2505.23737, 2025.
>
> [3]. Convergence analysis of adaptive gradient methods under refined smoothness and noise assumptions. COLT 2025.
>
> $\textbf{Weakness 3 (also to Weakness 2 from Reviewer qPF8)}$
>
> Condition (3) represents a trade‑off between theoretical guarantees and practical behavior. From the perspective of worst‑case theoretical bounds, (6) yields the optimal convergence rate, and condition (3) can be removed since it is always satisfied with $\hat\varepsilon=\varepsilon$. From a practical standpoint, (4) better explains why setting $\varepsilon$ to an extremely small value—as commonly done in practice—does not hinder fast convergence in real-world scenarios. For instance, Shi et al. (2023) adopt $\epsilon=10^{-12}$ in their implementation, and such small values do not degrade empirical training performance.
>
> $\textbf{Weakness 4 and Question 3}$
>
> Please see the response to $\textit{Weakness 3 from Reviewer qPF8}$.
>
> $\textbf{Question 1}$
>
> Assumption 3 is primarily adopted to facilitate the proof. Both inequalities in Assumption 3 are essential in the proof of Lemma 3.4 (see lines 300–303, left column) for establishing the tight convergence rate. We conjecture that the same convergence rate cannot be derived under the standard bounded variance assumption for the Shampoo-class methods.
>
> $\textbf{Question 4}$
>
> This condition is primarily introduced to facilitate the proof by bounding the update norm with a constant in Lemma 3.6. Recently, reference [4] observes that Adam performs near-optimally with $\theta=\beta$ in the LLM era, which aligns with our theoretical requirement. Thus, this condition also reflects the trends in practice.
>
> [4]. In Search of Adam's Secret Sauce. NeurIPS 2025.

---

> > ### Author Rebuttal · Reviewer_G8A9 · 2026-04-07
> >
> > I thank the authors for their response and maintain my original positive rating of the paper.

---

### Decision · Program_Chairs · 2026-04-30

**Decision:**

Accept (regular)

**Comment:**

The paper studies the convergence rate of AdamW-Style Shampoo. All reviewers recommend weak-acceptance. The reviewers agree that the paper considers an interesting problem and provides non-trivial and significant results. Reviewer LtwF raised some presentation issues that I encourage the authors to address in the revision. I recommend acceptance.